# PARROT: A Benchmark for Evaluating LLMs in Cross-System SQL Translation

**Wei Zhou**[1], **Guoliang Li**[2], **Haoyu Wang**[3], **Yuxing Han**[3], **Xufei Wu**[1], **Fan Wu**[1],
**Xuanhe Zhou** ✉[1]

[1] Shanghai Jiao Tong University [2] Tsinghua University [3] ByteDance
weizhoudb@sjtu.edu.cn

## Abstract

Large language models (LLMs) have shown increasing effectiveness in Text-to-SQL tasks. However, another closely related problem, Cross-System SQL Translation (a.k.a., SQL-to-SQL), which adapts a query written for one database system (e.g., MySQL) into its equivalent one for another system (e.g., Click-House), is of great practical importance but remains underexplored. Existing SQL benchmarks are not well-suited for SQL-to-SQL evaluation, which (1) focus on a limited set of database systems (often just SQLite) and (2) cannot capture many system-specific SQL dialects (e.g., customized functions, data types, and syntax rules). Thus, in this paper, we introduce **PARROT**, a **P**ractical **A**nd **R**ealistic Benchma**R**k for Cr**O**ss-System SQL **T**ranslation. PARROT comprises 598 translation pairs from 38 open-source benchmarks and real-world business services, specifically prepared to challenge system-specific SQL understanding (e.g., LLMs achieve lower than **38.53%** accuracy on average). We also provide multiple benchmark variants, including PARROT-Diverse with 28,003 translations (for extensive syntax testing) and PARROT-Simple with 5,306 representative samples (for focused stress testing), covering 22 production-grade database systems. To promote future research, we release a public leaderboard and source code at: `https://code4db.github.io/parrot-bench/`.

## 1 Introduction

Understanding and processing database SQL queries is a key criterion for evaluating large language models (LLMs) in both general and specific domains [1, 2]. However, existing researches mainly focus on advancing LLMs in the Text-to-SQL task [3]. In contrast, Cross-System SQL translation, so-called SQL-to-SQL, aims to adapt a SQL query written for one database system (e.g., MySQL) into an equivalent query for another system (e.g., ClickHouse), which is of critical practical importance in real-world scenarios, where enterprises frequently operate heterogeneous database environments and require seamless query migration across systems. Despite its significance, existing SQL benchmarks are mainly for Text-to-SQL, which are ill-suited for evaluating SQL-to-SQL capabilities. That is, they typically target a narrow range of database systems (mostly SQLite) and fail to capture diverse, system-specific dialect characteristics. As shown at the top of Figure 1, by testing with representative SQLs, we can identify critical problems of existing models in the SQL-to-SQL problem:

• **SQL-①** needs to be modified in calculation to execute in MySQL (i.e., "1 / col" → "1 / NULLIF(col, 0)") to avoid division-by-zero errors. However, the tested LLM (GPT-4o) fails to inject this safeguard because it lacks dialect-specific error-handling knowledge.

• **SQL-②** uses "GROUP BY ROLLUP($\cdots$)" that requires MySQL-specific syntax adjustments. GPT-4o cannot accurately locate and adapt the nested 'ROLLUP' due to distractions from lengthy unrelated clauses and an inability to isolate dialect-critical constructs.

39th Conference on Neural Information Processing Systems (NeurIPS 2025) Track on Datasets and Benchmarks.

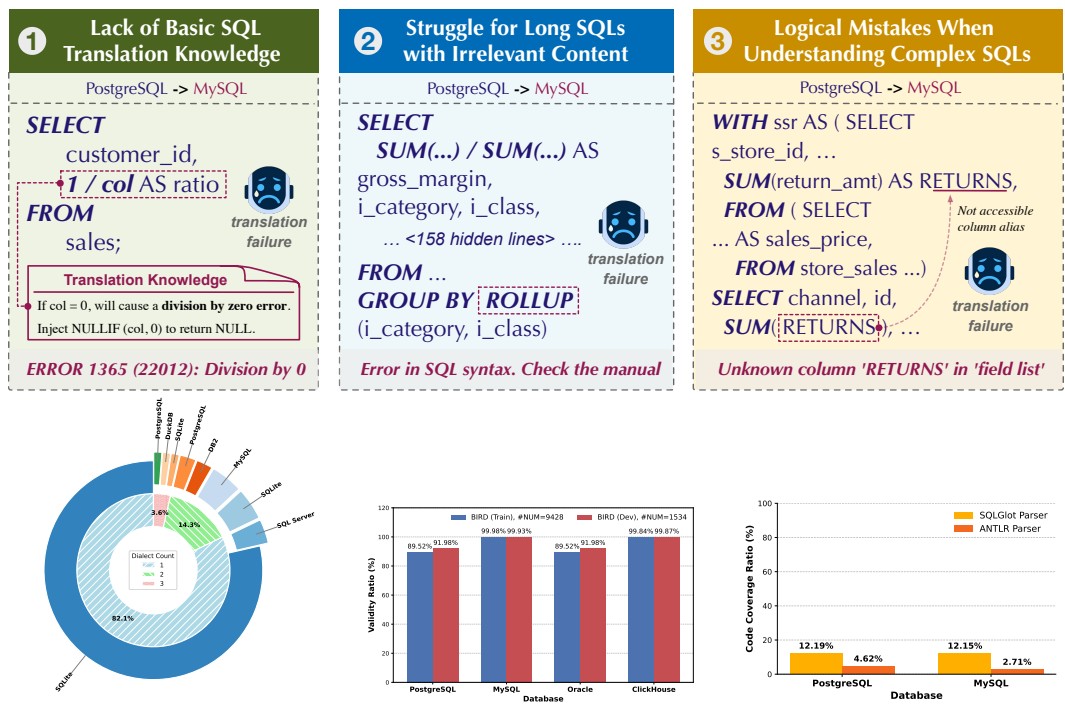

Figure 1: **Top** – Example queries illustrating key limitations of LLMs in SQL-to-SQL translation. **Bottom** – Empirical statistics from 28 open-source SQL-related benchmarks: (1) *Left:* Most benchmarks focus solely on SQLite (limited system diversity); (2) *Middle:* Over 89% of BIRD benchmark queries are system-agnostic (inadequate system coverage); (3) *Right:* Fewer than 13% of PostgreSQL and MySQL queries in BIRD-mini exhibit system-specific syntax (low dialect diversity).

● **SQL-③** defines an alias 'RETURNS' in a CTE subquery, which is not accessible in the outer query in MySQL. However, the LLM mistakenly assumes alias visibility across scopes, resulting in a reference to an undefined column and semantic failure during execution.

**Limitations of Existing Benchmarks.** Existing benchmarks lack sufficient such SQL queries for the SQL-to-SQL task. As shown in the bottom of Figure 1, our investigation of 28 open-source SQL-related benchmarks reveals several critical limitations. First, most benchmarks are designed for NL2SQL tasks and focus on a limited set of systems (e.g., primarily SQLite). Their queries are typically simple and do not require system-specific translation, making them unsuitable for this task. In contrast, real-world queries (e.g., those involving UDFs) often demand complex translation across database systems. Second, although a small portion of queries (e.g., fewer than 13% in BIRD-mini) require system-specific handling, they lack corresponding labels across multiple systems, offering only single-system SQLs, which limits their usability. Third, the volume of translation-relevant queries is small, and many critical SQL translation scenarios are underrepresented.

**Our Methodology.** To close this gap, we introduce **PARROT** (Practical And Realistic BenchmaRk for CrOss-System SQL Translation), the first large-scale dataset and evaluation suite dedicated to cross-system SQL translation. First, we curate a *diverse translation corpus* of 598 manually verified query pairs from 38 public benchmarks and real-world business applications, maximizing dialect diversity and real-world relevance. Second, we craft a *specialized challenge set* of 5,306 unit-style test cases spanning 22 production-grade database systems that isolate system-specific constructs (e.g., window-function variants, geo-types, and bitmap operations), thereby exposing brittle model behaviors invisible in prior work. Third, we provide an *augmented training pool* of 28,003 SQL statements mined and automatically tagged with dialect information. Fourth, we propose a *unified evaluation protocol* featuring reference executors, schema normalizers, and an execution-first metric that rewards semantic correctness over superficial string similarity. Finally, we release extensive *community resources*, including a public leaderboard, an open-sourced annotation toolchain, and two lighter benchmark variants, i.e., PARROT-DIVERSE for extensive syntax tests and PARROT-SIMPLE

for focused stress testing, and so researchers and practitioners can tailor evaluation to their specific needs. Empirical analysis reveals that state-of-the-art LLMs fail to achieve desirable performance across different dialects (i.e., ranging from around 17% - 60%), underscoring substantial headroom for future research.

## 2 Problem Formulation

***Cross-System SQL Translation*** is the task of converting a SQL query in a source database system (e.g., PostgreSQL) into a form that (1) strictly conforms to the target system's SQL *syntax* and (2) preserves the original query's *semantics*, so that it executes with *equivalent functionality* on the target database system (e.g., ClickHouse).

**Functional Equivalence.** The functional equivalence requires two query operations to be both syntactically compatible and semantically consistent. A query operation $q_i^T$, which is an implementation of syntax $S_i^T$, in database $D^T$ is *functionally equivalent* to a query operation $q_i^S$ in database $D^S$ if it adheres to the syntax standards in $D^S$ (i.e., *syntactically compatible*) and produces the same execution results or has the same effect as $q_i^S$ (i.e., *semantically consistent*).

For example, in PostgreSQL, the function CURRENT_TIMESTAMP returns the current date and time, while in MySQL, the equivalent function is NOW(). These operations are functionally equivalent, both producing the current system timestamp, although their syntax (dialects) differ.

**Cross-System SQL Translation.** Given a query $Q^S$ written in a source system SQL, $Q^S$ is composed of one or more operations $\{q_i^S\}$. *Cross-System SQL Translation* refers to the process of mapping each operation $q_i^S$ to one or more functionally equivalent operations in the target system SQL. The translated query $Q^T$ must (1) strictly follow the target dialect syntax $S^T$ (i.e., syntactically compatible with no runtime errors) and (2) maintain functional equivalence to $Q^S$ (i.e., semantically consistent to produce the same results). We utilize dialect to refer to SQLs designed for specific data systems.

Cross-System SQL translation (a.k.a, SQL-to-SQL) is a fundamental task that can serve as a foundation for a wide range of applications (e.g., data transformation pipelines, object-relational mapping tools, and cross-database interoperability), including text-to-SQL by translating their output (often tailored for SQLite) automatically into production-ready dialects used across different enterprise systems. For instance, at ByteDance, auto-generated queries from business intelligence tools require translation when migrating from Hive or Spark to ClickHouse database to ensure performance and compatibility. This process sustains a production workload exceeding 1,000 queries per second (QPS) while adapting syntax, optimizations, and engine-specific functions.

## 3 Collection and Curation of PARROT

PARROT is constructed using real-world SQLs for two key reasons: (1) Assembling representative workloads by humans is both labor-intensive and requires insightful domain expertise; (2) Although LLM-based query synthesis enables large-scale generation, the resulting queries often lack the structural nuances and operational patterns characteristic of production workloads (e.g., complex nested structures for specific service SQLs), making them less effective in reflecting real scenarios [4].

Overall, we first collect SQL samples from public open-source repositories as well as private proprietary workloads. The collected queries then pass through a rigorous curation pipeline (including clustering the SQLs based on their normalized representation and selecting the representative ones) that retains only those queries satisfying Jim Gray's four benchmark design principles [5].

### 3.1 SQL Source Collection

To make the prepared benchmark practical and realistic, we collect real-world queries from both the open-source and private domains rather than synthesizing from scratch.

• **Open-Source Domain.** To make the benchmark collection more practical, we collect SQLs available online in two ways: *(1) Open-Source Benchmark:* the dataset for benchmarking SQL-related tasks, including NL2SQL benchmarks [6, 7, 8, 9] for natural language interface and database specialized benchmarks [10, 11] for dedicated query optimization. Specifically, we collect the SQLs

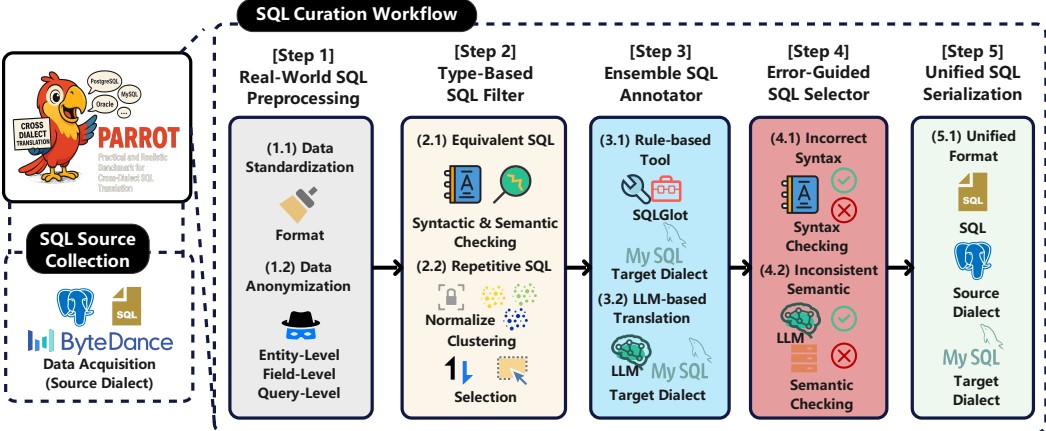

Figure 2: SQL Source Collection and Curation Workflow of **PARROT**.

from 38 benchmarks; *(2) Public Code Repository:* the hosting platform of actively maintained translation tools (e.g., SQLGlot [12], jOOQ [13]), including the test cases involved in the code repositories and the queries from the relevant GitHub issues (e.g., the ones with the keywords of "translation"). Specifically, we collect 1,041 tesecases from the repository.

• **Private Proprietary Domain.** To make the benchmark more realistic, we further introduce a dataset that includes real-world SQLs derived from ByteDance's internal data business scenarios. It encompasses 102 tables and comprises 343 SQL pairs. ByteDance has independently developed the cloud-native data warehouse system, ByteHouse [4], which adheres to ClickHouse syntax. During the process of migrating existing OLAP services within the company, a significant number of SQL queries written in Postgres-variant syntax needed to be rewritten into ClickHouse syntax. This dataset represents only a portion of the internal data. It was carefully created through manual rewriting and subsequent verification by senior SQL experts.

More details about SQL sources, including the collected domain analysis, are presented in Section A.

## 3.2 SQL Curation Workflow

However, the collected SQL queries require further refinement to serve as a qualified benchmark for cross-system translation, due to the following limitations.

**1. Redundancy and Limited Complexity:** Many queries are either duplicated from the same underlying templates (e.g., with different parameter values) or rely on simple, commonly used operations such as the SUM() and COUNT() aggregation functions. This lack of diversity and complexity limits the effectiveness of evaluation, as these translation-friendly queries fail to reflect the syntactic and semantic challenges present in real-world scenarios.

**2. Single-Dialect Limitation:** Since existing benchmarks were not designed for dialect translation, the majority of queries are written in a single SQL dialect (one database system). Consequently, additional annotation and mapping are necessary to construct functionally equivalent queries in other dialects, enabling effective cross-dialect evaluation.

To address these issues, **PARROT** proposes a comprehensive SQL curation workflow dedicated to the translation benchmark construction. As shown in Figure 2, it consists of five steps.

• **Step 1: Real-World SQL Preprocessing.** Since SQLs from different benchmarks are typically structured in a heterogeneous format, we first integrate these SQLs into a standardized representation to facilitate the subsequent steps. Specifically, we collect SQLs from the domains mentioned above, format these SQLs (e.g., remove redundant whitespace) and deduplicate the repetitive ones, where each line corresponds to a single SQL. Moreover, to protect privacy in benchmarks derived from proprietary domains, we apply three levels of anonymization.

*(1) Entity-level Anonymization:* Obscure schema semantics by replacing descriptive table and column names with generic identifiers (e.g., `table_1`, `column_1`) and randomly merging tables based on join relationships to mask the original schema structure.

*(2) Field-level Anonymization:* Protect sensitive data in the field content by injecting noise into numeric fields and substituting text fields with synthetic or placeholder values (e.g., `NULL`), while preserving data utility, as specific values typically do not affect cross-system translation.

*(3) Query-level Anonymization:* Remove identifiable query patterns by abstracting structural elements such as continuous identical filter conditions. The redundant snippets are pruned to generalize the query form while maintaining its syntactic integrity and logical flow.

• **Step 2: Type-Based SQL Filter.** To eliminate low-quality SQL queries from the large corpus and reduce the burden of subsequent steps, we propose automated filtering strategies tailored to address different types of deficiencies in the collected SQLs.

*(1) Syntax and Semantic Checking for Equivalent SQLs:* Given that some of the integrated SQLs might already be equivalent in the target systems, wastes over assessing these SQLs should be prevented. Therefore, we first utilize parsers with dialect syntax (e.g., the ANTLR) to exclude SQLs that are already compatible (i.e., no parsing error raised).

*(2) Clustering then Selection for Repetitive SQLs:* Based on the observation that queries originating from the same query template (i.e., only differ in the parameters) occupy a large proportion, we employ clustering then selection for this problem. First, we normalize the SQLs and propose a prefix-based method to cluster them into several groups. Specifically, we normalize the identifiers in the SQLs (e.g., replace specific table and column names with the unified "`table`" and "`column`" representation). Moreover, to enhance the clustering accuracy, we shrink multiple identifiers into a single one representation (e.g., transform continuous "`table, table, table`" into a single "`table`"). With a specified prefix length proportional to the original SQL length (e.g., 0.25), we cluster SQLs of the same prefix into the same groups. Second, we select the SQLs based on the clustered groups and utilize a code coverage assessment tool to enrich the diversity. We sort the SQLs in the descending order over the average SQL length within one group with the intuition that longer SQLs are typically more complex and diverse. Then, we successively sample one SQL from the current group and invoke the coverage assessment tool to determine whether it can increase the code coverage of the parser. If so, the corresponding SQL is added to an unique set for later processing. We proceed to the next group if the sample SQLs fail to increase the coverage within specified rounds (e.g., 5) and the whole process terminates for the last group.

• **Step 3: Ensemble SQL Annotator.** Given that existing benchmark only provides SQLs within single dialect, we introduce an automatic annotation mechanism to effectively expand these SQLs to other dialects. Specifically, we utilize the traditional rule-based tools to derive the initial annotations (which will be validated in later steps). Considering different methods might vary in the effectiveness across different dialects [14], we adopt an ensemble paradigm to enhance the annotation accuracy. We employ multiple tools (i.e., SQLGlot [12], jOOQ [13]) for translation and accumulate their results. We also consider a recent LLM-based method as the candidate annotator [14]. However, we prioritize the rule-based tools considering their efficiency and effectiveness over the collected diverse SQLs of a large volume. Besides, we also employ small-scale LLMs (e.g., Llama3.1-8B [15]) as the annotator and grounded as the baseline for a guidance of later selection. These annotated SQLs are then serve as the input to the next step for validation and selection.

• **Step 4: Error-Guided SQL Selector.** The translation tools might inevitably produce incorrect translations (e.g., missing specific rules), introducing errors in the constructed benchmark. Therefore, we further employ a hybrid strategy to select and revise the annotated SQLs based on the possible error types. Overall, the translation errors can be classified into two categories, tightly coupled with the characteristics of this problem introduced in Section 2.

*(1) Incorrect Syntax:* We rely on parsers with dialect syntax (e.g., the ANTLR [16]) to verify whether the annotated SQLs violate dialect-specific syntax. For SQLs that raise parsing errors, we call for human experts (e.g., ByteHouse engineers) to fix these errors. The human experts collaborate with LLMs (e.g., provide related hints as assistants) in the fixing process to enhance both the accuracy and the efficiency. The revised SQLs are passed to the same parsers to check if any syntax errors persist.

Table 1: Statistics of Different Datasets in **PARROT**.

| Dataset | #Dialect | #SQL | #Token / SQL | | | #Translation Type |
|---|---|---|---|---|---|---|
| | | | 25th | Medium | 75th | |
| **PARROT** | 8 | 598 | 75.0 | 249.0 | 951.0 | 7 |
| **PARROT**-Diverse | 22 | 28,003 | 29.0 | 47.0 | 71.0 | 7 |
| **PARROT**-Simple | 22 | 5,306 | 4.0 | 6.0 | 10.0 | 7 |

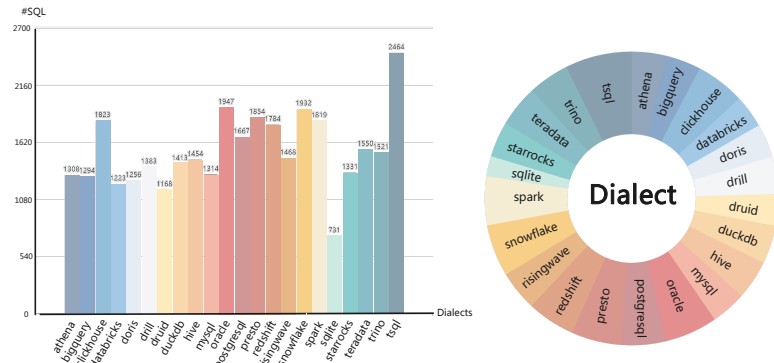

Figure 3: SQL Distribution over Different Dialects in **PARROT**.

If the syntax errors can be not resolved within given attempts, the SQLs will be excluded in the benchmark. In contrast, the syntactically-correct SQLs undergo the subsequent semantic checking.

*(2) Inconsistent Semantic:* Total reliance on human experts to perform semantic checking over SQLs of large volume is impractical. Hence, we propose an automatic strategy to determine the equivalence based on the execution results of the generated testcases. Recent studies have shown that LLMs have the capability to generate effective testcases, thus we also utilize LLMs for testcase generation. Specifically, we carefully prompt LLMs to generate SQLs (i.e., the INSERT statements) that ensure non-empty execution results of the two SQLs. We also specify to steer LLMs to generate SQLs that can lead to inconsistent results in the instructions. This generation process takes place within given rounds (e.g., 5) and the SQLs are excluded from the final benchmark once inconsistent result occurs. Besides, we also remove the SQLs if they can be already successfully translated by the small-scale LLMs in the last step to enhance the difficulty of the constructed benchmark.

• **Step 5: Unified SQL Serialization.** With the above processing steps, we finally dump the benchmark into a unified ".json" file. As shown in Figure 2, each item in the file corresponds to a SQL pair including the specification of the unique data id, the source dialect, the target dialect, and the corresponding SQLs.

## 4 PARROT Benchmark Analysis

We present more details about how **PARROT** meets with the benchmark design criteria proposed by Jim Gray [5] and showcase detailed information about the underlying benchmark statistics.

• **Relevance.** **PARROT** is the first benchmark for assessing LLMs in dialect translation, including a collection of 33,952 SQL pairs across 22 data systems. It accumulates the real-world SQLs from both open-source domains and private domains, including 38 SQL-relevant benchmarks and enterprise customer workloads encompassing 102 tables in ByteHouse business scenarios. Furthermore, these SQLs vary in the intrinsic complexity (e.g., the token length can up to 2,182 tokens) and the translation difficulty (i.e., involve multiple translation types introduced in Section 2).

• **Scalability.** **PARROT** offers several variants with additional expanded datasets to satisfy the assessment purposes in diverse scenarios. Apart from the main dataset, it provides three variants.

Table 2: Translation Accuracy (%) over **PARROT** across Diverse Dialects (* $\rightarrow$ PostgreSQL indicates PostgreSQL serves as the target dialect in the translation process).

| Model | * ↓ PostgreSQL | * ↓ MySQL | * ↓ Oracle | * ↓ DuckDB | * ↓ SQL Server |
|---|---|---|---|---|---|
| **Open-Source LLM** | | | | | |
| DeepSeek-R1 7B | 17.24 | 20.59 | 17.24 | 14.29 | 15.79 |
| DeepSeek-R1 32B | **58.62** | **58.82** | 39.66 | 10.71 | **42.11** |
| DeepSeek-Coder-V2 Lite | 34.48 | 32.35 | 32.76 | 3.57 | 21.05 |
| DeepSeek V3 671B | 55.17 | 55.88 | 51.72 | 53.57 | 36.84 |
| DeepSeek R1 671B | 48.28 | 44.12 | 50.00 | 42.86 | 36.84 |
| **Proprietary LLM** | | | | | |
| GPT-4o | **58.62** | 50.00 | 55.17 | **60.71** | **42.11** |
| o3-mini | 31.03 | 8.82 | 43.10 | 35.71 | 21.05 |
| Gemini 2.5 Pro | 48.28 | 23.53 | 32.76 | 53.57 | 21.05 |
| Claude 3.7 Sonnet | **58.62** | 44.12 | **58.00** | 42.86 | 36.84 |

**(1) PARROT-DIVERSE:** It consists 28,003 samples of SQL pairs across 22 dialects. It is aimed at the evaluation of LLMs across diverse data systems and can measure whether LLMs perform equivalent well among the data systems (i.e., obtain superior translation performance).

**(2) PARROT-SIMPLE:** It consists of 5,306 SQL pairs based on testcases collected from the code repository of rule-based translation tools. The testcases are typically SQL snippets dedicated to a single translation type. Therefore, this variant can be utilize to measure whether LLMs internalize specific translations.

• **Simplicity.** **PARROT** selects representative SQL queries from diverse domains while avoiding redundancy that could compromise evaluation efficiency. As outlined in Section 3, it follows a systematic SQL collection and curation workflow to prepare high-quality benchmark queries. This process significantly reduces the volume of raw SQLs, e.g., distilling 9,912,231 SQL pairs down to 28,003 representative queries, by identifying and retaining only those that are structurally and semantically diverse within defined groups.

• **Portability.** **PARROT** proposes multiple assessment strategy in terms of different aspects to enable it adapt to diverse setting and evaluation scenarios. Specifically, it currently supports the following assessment criteria corresponding to two aspects (i.e., syntax and semantic) in functional equivalence defined in Section 2.

**(1) Dialect Compatability ($Acc_{EX}$):** The ratio of the translated queries that are executable (i.e., syntactically correct) in the target database without raising incompatibility error (e.g., incorrect data types or functions);

**(2) Result Consistency ($Acc_{RES}$):** The ratio of the translated queries that return the strictly identical results (i.e., semantically consistent) in the target database as the source queries in the source database, including the returned data format, precision, and displayed order.

The SQLs which require translation are typically tightly coupled with the daily business service. Hence, their execution efficiency is also an important factor, where we can also propose an relevant efficiency score [17]. However, the efficiency can be enhanced by subsequent utilization of external tools [18], our primary focus lies in the translation accuracy in this paper.

## 5 Experiments

### 5.1 Experimental Setup

**Baselines.** We assess the translation performance of prevalent LLMs in terms of three aspects in the experiments. *(1) Usage License:* We consider both the open-source LLMs (e.g., DeepSeek-V3 671B [19]) and the proprietary LLMs (e.g., Claude 3.7 Sonnet and GPT-4o [20]); *(2) Parameter Scale:* We consider LLMs with varied and increasing parameter scales (e.g., from DeepSeek-R1 7B [21] to o3-mini and o1-preview); *(3) Task Scope:* We consider both LLMs that can handle diverse

Table 3: Translation Accuracy (%) with Real-World Workload in ByteHouse.

| Model | $Acc_{EX}$ | $Acc_{RES}$ |
|---|---|---|
| **Open-Source LLM** | | |
| **DeepSeek-R1 32B** | 21.00 | 16.91 |
| **DeepSeek-V3 671B** | 39.94 | 32.65 |
| **DeepSeek-R1 671B** | 46.94 | 40.52 |
| **Proprietary LLM** | | |
| **GPT-4o** | 23.91 | 21.87 |
| **o3-mini** | **58.60** | **54.23** |
| **o1-preview** | 56.26 | 48.69 |
| **Claude 3.7 Sonnet** | 24.20 | 22.74 |

Table 4: Efficiency Analysis ($\frac{Time_{target}}{Time_{source}}$) with Real-World Workload in ByteHouse.

| Model | Mean | Median | 95th | 99th |
|---|---|---|---|---|
| **Open-Source LLM** | | | | |
| **DeepSeek-R1 32B** | 0.59 | **0.49** | 1.18 | 1.32 |
| **DeepSeek-V3 671B** | 0.62 | 0.53 | 1.13 | **1.25** |
| **DeepSeek-R1 671B** | 0.58 | 0.51 | **1.11** | 1.38 |
| **Proprietary LLM** | | | | |
| **GPT-4o** | 0.63 | 0.50 | 1.33 | 1.44 |
| **o3-mini** | 0.59 | 0.53 | 1.12 | 1.38 |
| **o1-preview** | **0.57** | 0.50 | 1.15 | 1.37 |
| **Claude 3.7 Sonnet** | 0.62 | 0.51 | 1.20 | 1.33 |

tasks with a general purpose (e.g., DeepSeek-R1 671B [21]) and dedicated to specialized code-related tasks (e.g., DeepSeek-Coder-V2 Lite). Each LLM performs dialect translations based on the well-crafted prompt including detailed problem instructions that can be found at Section A.

**Evaluation**. We adopt the evaluation metrics (i.e., $Acc_{EX}$ and $Acc_{RES}$) defined in Section 4. The workstation setup is two Intel(R) Xeon(R) CPU E5-2678 v3 @ 2.50GHz, 256 GB main memory, and four GeForce RTX 3080 and H100 Ti graphics cards.

## 5.2 Comparative Analysis

We assess the translation performance across diverse dialects of different LLMs over **PARROT**.

*(Observation 1) -* LLMs *exhibit performance oscillation across the translations among different dialects.* As shown in Table 2, we notice that LLMs showcase different capabilities over the evaluated dialects. Specifically, GPT-4o achieves the highest accuracy (i.e., 58.62%) over the translation to PostgreSQL while its performance degrades with the accuracy (i.e., 50.00%) over the translation to MySQL, even lower than DeepSeek-R1 32B. It corresponds to the characteristics of the dialect translation problem, which involves a collection of stringent syntax standards among different dialects. Therefore, LLMs are expected to clearly capture the nuanced differences of diverse dialect standards to perform well. This phenomenon makes us reflect upon how to design a LLM or augment existing ones to specifically enhance the dialect translation capability so that different dialects can be equivalently handled well. Moreover, the capability can only be obtained to develop specialized LLM for each or similar dialect pairs.

*(Observation 2) - A larger scale of the parameter volume might not contribute to the consistent improvement of the translation accuracy.* As displayed in Table 2, we observe that large scale or more advanced LLMs might not perform better than the smaller ones. For example, DeepSeek-R1 32B performs better over translations to PostgreSQL, MySQL, and SQL Server with the respective accuracy 58.62%, 58.82%, and 42.11% than 48.28%, 44.12%, and 36.84% by DeepSeek-R1 671B. Moreover, to our surprise, we notice that advanced reasoning LLMs (i.e., o3-mini) exhibit undesirable translation performance. This result reflects the mismatched capability enhancement of large scale or advanced LLMs, typically aimed at complex problems with intrincate reasoning process unlike the capability required in cross-system dialect translation. Based on the experimental results, we identify two abilities are desired for accurate translation: (1) the SQL understanding ability to analyze and write specific SQLs and (2) the SQL syntax matching ability to be aware of the equivalent operations.

*(Observation 3) -* LLMs *struggle to obtain accurate translation when the SQLs become more lengthy with more complex operations.* We present a more fine-grained analysis about the translation performance of LLMs considering the characteristics of input SQLs. Specifically, we tokenize the SQLs and classify them into several groups based on the number of derived tokens. As shown in Table 3 and Figure 3, we observe that all the LLMs encounter performance regression when the SQLs become more lengthy despite the translated queries do not exhibit degraded performance (e.g., no extreme slowdowns or timeouts in Table 4). Specifically, all the LLMs exhibit an average performance degration when the number of tokens involved in the SQL increase from $0 - 402$ to $1214 - 2182$. This result can be attributed to two aspects: (1) longer queries typically involve more operations to be resolved, thus increasing the translation difficulty; (2) lengthy queries increase the

Table 5: Error Analysis over **PARROT** with Real-World Workload in ByteHouse.

| Model | DeepSeek-R1 32B | DeepSeek V3 671B | DeepSeek-R1 671B | GPT-4o | o3-mini | o1-preview | Claude 3.7 Sonnet |
|---|---|---|---|---|---|---|---|
| Syntax Parsing | 0.05 | **0.28** | 0.10 | 0.00 | 0.11 | 0.03 | 0.03 |
| Identifier Resolution | 0.02 | 0.00 | 0.07 | 0.00 | 0.01 | 0.06 | 0.00 |
| Function Resolution | **0.28** | 0.00 | 0.05 | **0.40** | **0.64** | **0.37** | 0.04 |
| Function Usage | **0.62** | 0.72 | 0.70 | 0.60 | 0.24 | 0.54 | 0.93 |
| Type Compatibility | 0.01 | 0.00 | 0.08 | 0.00 | 0.00 | 0.00 | 0.00 |
| Other Errors | 0.02 | 0.00 | 0.00 | 0.00 | 0.00 | 0.00 | 0.00 |

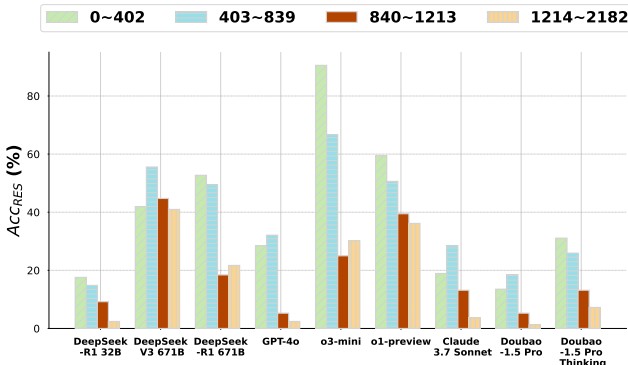

Figure 4: Distribution of Translation Accuracy (%) over SQL Length.

risk of triggering the limitation of LLMs, including the hallucination and lost-in-the-middle problem. Therefore, it calls for techniques to enable LLMs perform accurate translation over lengthy SQLs (e.g., the segment-based translation strategy proposed in [22, 14]).

## 5.3 Error Analysis

We further conducted a detailed analysis of failure cases over the results in real-world ByteHouse workloads, including error frequency distributions across different LLMs. These failures fall into six categories. (1) Syntax Parsing: SQL statements that violate grammar rules, e.g., missing clauses or invalid keyword usage. (2) Identifier Resolution: References to undefined or ambiguous database objects, e.g., using a column name when multiple tables share the same field. (3) Function Resolution: Use of unknown or misclassified functions, e.g., calling a non-existent function or incorrectly treating a scalar function as an aggregate. (4) Function Usage: Valid functions called with incorrect argument types, counts, or values, e.g., passing a string where a number is expected. (5) Type Compatibility: Operations that combine incompatible data types or schemas, e.g., performing UNION ALL between two queries with different returned value types. (6) Other Errors: Miscellaneous issues outside the above categories, such as unsupported SQL features or incomplete syntax constructs.

From Table 5, we have two observations. *(1) Failure patterns vary across different LLMs.* For instance, DeepSeek-V3 671B primarily struggles with Function Usage (0.72) and Syntax Parsing (0.28), indicating difficulties in correctly applying functions and adhering to SQL grammar. In contrast, o1-preview exhibits a higher rate of failures in Function Resolution (0.37) and Function Usage (0.54), suggesting challenges in identifying correct functions and using them properly. *(2) The majority of failures stem from misuse of built-in functions.* For example, Function Resolution and Function Usage account for 89% of all errors on average, highlighting that the translation of database-specific functions is the most significant obstacle. These observations align with the complexity posed by the diverse and extensive range of custom functions implemented across databases, which significantly increases the difficulty for LLMs to accurately translate functions across dialects.

## 5.4 Case Study

We perform a case study based on the detailed analysis of an SQL query that failed to be translated by LLMs. As shown in Table 6, this SQL is extracted from the ByteHouse real-world customer workloads, where LLMs (e.g., o3-mini) incur two translation errors. The first error involves the

Table 6: Case Study of Translation Errors Incurred by LLM.

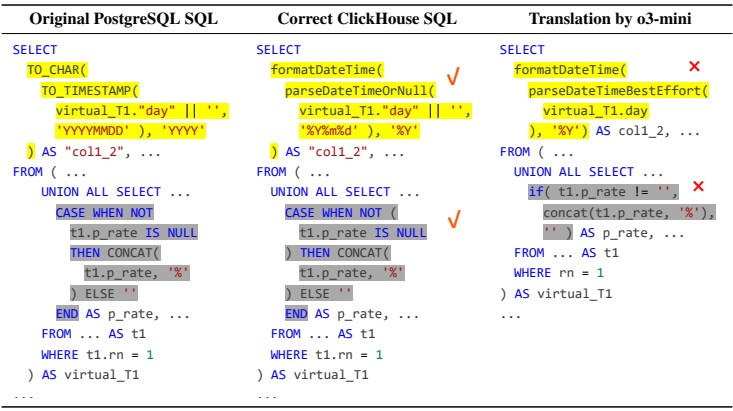

translation over operation that intends to convert the input string into a datetime data type. Specifically, the source PostgreSQL-variant SQL operation (i.e., `TO_TIMESTAMP(virtual_T1."day" || '', 'YYYYMMDD'))` converts the column values (i.e., `virtual_T1."day"`) into a time stamp data type based on the specified format (i.e., `'YYYYMMDD'`). Since the column (i.e., `virtual_T1."day"`) is defined as an integer data type, it utilizes an additional expression (i.e., `|| ''`) to transform it into a string data type so that it can be processed by the `TO_TIMESTAMP()` function. However, o3-mini directly translates this operation into `parseDateTimeBestEffort(virtual_T1.day)` in ByteHouse, where the column (i.e., `virtual_T1.day`) is not converted to an integer data type and leads to runtime errors (i.e., `Illegal type Int64 of first argument of function parseDateTimeBestEffort`). Moreover, the datetime format equivalent to `'YYYYMMDD'` in the source SQL is left out. The second error refers to the incorrect processing of columns with NULL values. Specifically, the `CASE WHEN NOT t1.p_rate IS NULL THEN CONCAT(t1.p_rate, '%') ELSE '' END` in the source PostgreSQL-variant SQL processes `t1.p_rate` with different logics (i.e., `CASE WHEN`) by validating whether it corresponds to NULL values with `IS NOT NULL` operation. However, o3-mini incorrectly translates the validation over the NULL values to `!= ''` and leads to a runtime error (i.e., `Cannot read floating point value: while converting '' to Float64`). Based on these error analyses, we notice that even though LLMs can identify certain equivalent translations with internal knowledge (e.g., `TO_TIMESTAMP()` and `parseDateTimeBestEffort()` functions), they are still too careless to miss some operations in the source SQLs and struggle to ensure the consistency over stringent dialect syntax standards.

## 6 Related Work

**Dialect Translation Tools.** Tools such as SQLGlot [12], SQLines [23], and jOOQ [13] support rule-based translation across dialects. These systems typically encode translation logic through handcrafted rules or pattern-based templates, enabling basic conversion of common syntax.

**NL2SQL Benchmarks.** Benchmarks such as Spider [8], BIRD [17], and WikiSQL [9] have significantly advanced NL2SQL research by providing large-scale datasets of natural language questions paired with SQL queries. However, these datasets primarily target a single SQL dialect (most commonly SQLite) and do not reflect the syntactic or semantic variations across database systems. For instance, they lack annotations indicating dialect-specific syntax. This limits the applicability of existing NL2SQL benchmarks to the problem of cross-system SQL translation, where both syntactic fidelity and semantic correctness must be preserved across diverse systems.

## 7 Conclusion

In this paper, we propose PARROT, which is the first benchmark for effectively evaluating cross-system SQL translation. Through a carefully curated and richly diverse dataset, specialized diagnostic cases, and a robust evaluation protocol, PARROT enables a comprehensive and practical assessment of existing LLMs in system-specific translation. Our benchmark not only facilitates reproducible research but also empowers the development of more robust, accurate, and generalizable SQL translation methods across different database systems.

## Acknowledgments and Disclosure of Funding

Xuanhe Zhou is the corresponding author. This work was supported in part by National Key R&D Program of China (No. 2023YFB4502400), in part by China NSF grant (No. 62441236, 62372296, 62432007, U25A6024, U25A20437, 62525202, 62232009, 62502304), in part by Fundamental and Interdisciplinary Disciplines Breakthrough Plan of the Ministry of Education of China (No. JYB2025XDXM103), in part by Alibaba Group through Alibaba Innovation Research Program, in part by Tencent Rhino Bird Key Research Project, Shenzhen Project (CJGJZD20230724093403007), Zhongguancun Lab, and Beijing National Research Center for Information Science and Technology (BNRist), CCF Populus Grove Fund, ByteDance, Tencent, Ant Group through CCF-Ant Research Fund, Shanghai Jiao Tong University AI for Engineering Initiative.

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

# A  Technical Appendices and Supplementary Material

**PARROT** categorizes cross-dialect SQL translation challenges into several common types based on structural, lexical, and functional differences across database systems.

Table 7: Typically Translation Types in **PARROT**.

| Translation | Description |
|---|---|
| **Syntax Rule** | Differences in syntactic structure requirements across databases. |
| **Keyword** | Naming differences for reserved words or functional keywords. |
| **Data Type** | Naming or precision differences for equivalent logical data types. |
| **Operator & Built-in Function** | Name/behavior differences for operators or built-in functions. |
| **Stored Procedure** | Differences in definition and invocation syntax. |
| **UDF** | Differences in creation and usage of user-defined functions. |
| **Other** | Miscellaneous special differences (e.g., variable prefixes, comment symbols). |

Below, we present the details of the collected benchmarks included in **PARROT**, highlighting their sources, dialect coverage, and key statistics.

Table 8: Details of Collected Benchmarks in **PARROT**.

| Benchmark | Year | SQL Dialects Supported | Language | Domain Type | Turn | Collection |
|---|---|---|---|---|---|---|
| ATIS | 1994 | SQLite, MySQL | English | Single-domain | Single | Manual |
| GeoQuery | 1996 | MySQL, SQLite | English | Single-domain | Single | Manual |
| Restaurants | 2000 | SQLite | English | Single-domain | Single | Manual |
| Academic | 2014 | *Unspecified* | English | Single-domain | Single | Manual |
| IMDb | 2017 | *Unspecified* | English | Single-domain | Single | Manual |
| Yelp | 2017 | *Unspecified* | English | Single-domain | Single | Manual |
| Scholar | 2017 | *Unspecified* | English | Single-domain | Single | Manual |
| WikiSQL | 2017 | SQLite3 | English | Cross-domain | Single | Manual |
| Advising | 2018 | SQLite, MySQL | English | Single-domain | Single | Manual |
| Spider | 2018 | SQLite | English | Cross-domain | Single | Manual |
| SParC | 2019 | SQLite | English | Cross-domain | Multiple | Manual |
| CoSQL | 2019 | SQLite | English | Cross-domain | Multiple | Manual |
| CSpider | 2019 | SQLite | Chinese | Cross-domain | Single | Manual |
| MIMICSQL | 2020 | SQLite | English | Single-domain | Single | Hybrid† |
| SQUALL | 2020 | SQLite | English | Cross-domain | Single | Manual |
| FIBEN | 2020 | Db2, PostgreSQL | English | Single-domain | Single | Manual |
| ViText2SQL | 2020 | General SQL | Vietnamese | Cross-domain | Single | Manual |
| DuSQL | 2020 | *Unspecified* | Chinese | Cross-domain | Single | Hybrid† |
| PortugueseSpider | 2021 | SQLite | Portuguese | Cross-domain | Single | Hybrid† |
| CHASE | 2021 | SQLite | Chinese | Cross-domain | Multiple | Manual |
| Spider-Syn | 2021 | SQLite | English | Cross-domain | Single | Manual |
| Spider-DK | 2021 | SQLite | English | Cross-domain | Single | Manual |
| Spider-Realistic | 2021 | SQLite | English | Cross-domain | Single | Manual |
| KaggleDBQA | 2021 | SQLite | English | Cross-domain | Single | Manual |
| SEDE | 2021 | T-SQL | English | Single-domain | Single | Manual |
| MT-TEQL | 2021 | SQLite | English | Cross-domain | Single | Automatic |
| PAUQ | 2022 | SQLite | Russian | Cross-domain | Single | Manual |

| Benchmark | Year | SQL Dialects Supported | Language | Domain Type | Turn | Collection |
|---|---|---|---|---|---|---|
| knowSQL | 2022 | *Unspecified* | Chinese | Cross-domain | Single | Manual |
| Dr.Spider | 2023 | SQLite | English | Cross-domain | Single | Hybrid[†] |
| BIRD | 2023 | SQLite | English | Cross-domain | Single | Manual |
| AmbiQT | 2023 | SQLite | English | Cross-domain | Single | LLM-aided |
| ScienceBenchmark | 2024 | General SQL | English | Single-domain | Single | Hybrid[†] |
| BookSQL | 2024 | SQLite | English | Single-domain | Single | Manual |
| Archer | 2024 | SQLite | English/Chinese | Cross-domain | Single | Manual |
| BULL | 2024 | SQLite | English/Chinese | Single-domain | Single | Manual |
| Spider2 | 2024 | SQLite, DuckDB, PostgreSQL | English | Cross-domain | Single | Manual |
| TPC-H FROID | 2018 | T-SQL, PostgreSQL | English | Cross-domain | Single | Hybrid[†] |
| DSB | 2021 | T-SQL, PostgreSQL | English | Decision Support | Single | Hybrid[†] |
| TPC-DS | 2005 | T-SQL, PostgreSQL | English | Decision Support | Single | Hybrid[†] |
| SQL-ProcBench | 2021 | SQL Server, PostgreSQL, IBM Db2 | English | Enterprise workloads | Single | Production-derived |

[†] **Hybrid** means the dataset was created using both automatic generation and manual annotation.

We introduce the SQL annotation interface and prompt design adopted in **PARROT**, which facilitate efficient user interaction and enhance LLM-guided SQL understanding.

Table 9: SQL Annotation System and User Prompt in **PARROT**.

| **System Prompt** |
|---|

## CONTEXT ##
You are a database expert specializing in various SQL dialects, such as **{src_dialect}** and **{tgt_dialect}**, with a focus on accurately translating SQL queries between these dialects.

## OBJECTIVE ##
Your task is to translate the input SQL from **{src_dialect}** into **{tgt_dialect}**, ensuring the following criteria are met:
1. **Grammar Compliance**: The translated SQL must strictly adheres to the grammar and conventions of {tgt_dialect} (e.g., correct usage of keywords and functions);
2. **Functional Consistency**: The translated SQL should produce the same results and maintain the same functionality as the input SQL (e.g., same columns and data types).
3. **Clarity and Efficiency**: The translation should be clear and efficient, avoiding unnecessary complexity while achieving the same outcome.

During your translation, please consider the following candidate translation points:
1. **Keywords and Syntax**: Ensure {tgt_dialect} supports all the keywords from the input SQL, and that the syntax is correct;
2. **Built-In Functions**: Verify that any built-in functions from {src_dialect} are available in {tgt_dialect}, paying attention to the argument types and the return types;
3. **Data Types**: Ensure that {tgt_dialect} supports the data types used in the input SQL. Address any expressions that require explicit type conversions;

4. **Incompatibilities**: Resolve any other potential incompatibility issues during translation.

This task is crucial, and your successful translation will be recognized and rewarded.
Please start by carefully reviewing the input SQL and then proceed with the translation.

**User Prompt**

## INPUT ##
Please translate the input SQL from **{src_dialect}** to **{tgt_dialect}**.
The input SQL is:
```sql
{sql}
```
## OUTPUT FORMAT ##
Please return your response without any redundant information, strictly adhering to the
following format:
```json
{{
"Answer": "The translated SQL",
"Reasoning": "Your detailed reasoning for the translation steps (clear and succinct, no more
than 200 words)",
"Confidence": "The confidence score about your translation (0 - 1)"
}}
```

## OUTPUT ##

