# OpenReview forum: "PARROT: A Benchmark for Evaluating LLMs in Cross-System SQL Translation"
_NeurIPS.cc/2025/Datasets_and_Benchmarks_Track — NeurIPS 2025 Datasets and Benchmarks Track poster_

### Official Review · Reviewer_Hp1m · 2025-06-24

**Rating:** 4
**Confidence:** 4

**Summary:**

This paper presents PARROT, a benchmark designed to evaluate large language models (LLMs) in cross-system SQL translation (SQL-to-SQL). Unlike the widely studied Text-to-SQL task, SQL-to-SQL focuses on converting SQL queries from one database system dialect to another while preserving functionality. The authors curate the dataset from 38 public and proprietary sources (e.g., ByteDance), propose a five-stage curation workflow, and conduct comprehensive evaluations on multiple LLMs. PARROT includes variants (PARROT, PARROT-Diverse, PARROT-Simple) and introduces dual metrics for syntactic correctness and semantic fidelity.

**Dataset Code Accessibility:**

Yes

**Dataset Code Comments:**

The repository is well-organized and contains sufficient material to replicate the core results. The JSON-format serialization, translation type taxonomy, and sample prompts enhance usability for benchmarking and system development. The use of ANTLR and SQLGlot makes the pipeline technically robust and extensible.

**Ethical Considerations:**

No, there are no or only very minor ethics concerns

**Final Justification:**

The authors have addressed most of the reviewers' concerns. I hope that the authors will further review the relevant work in final version.

**Limitations Weaknesses:**

1. The case study is strong, but a taxonomy of common failure modes (e.g., date parsing, alias scope errors, aggregate translation) with frequencies could offer deeper insights.
2. Although the dataset is used to evaluate many LLMs, the authors provide limited discussion on how prompt engineering or decoding strategies impact results.
3. The Acc_RES metric assumes fully aligned schemas, but the paper lacks details on how schema differences, type coercion, or NULL handling are managed. In practice, cross-system translation often involves schema mismatches and differences in NULL semantics. Clarifying these aspects would improve reproducibility.

**Strengths Contributions:**

The paper addresses a practically important yet underexplored task—cross-dialect SQL translation—which is critical for enterprise-level database migration, system interoperability, and real-world automation.

- Novel benchmark focus: PARROT is the first benchmark specifically designed for SQL-to-SQL translation, filling a notable gap between Text-to-SQL and rule-based SQL conversion tools.

- Diverse, high-quality dataset: The authors compile 33,000+ SQL pairs spanning 22 dialects. The use of both open benchmarks and industrial SQLs (ByteHouse) adds realism and diversity.

- Rigorous curation pipeline: The 5-step processing pipeline (data standardization, clustering, ensemble annotation, error-guided filtering, serialization) demonstrates strong engineering effort and replicable methodology.

- Comprehensive experiments: Benchmarking across proprietary and open-source LLMs reveals critical performance gaps (e.g., hallucinations, inability to handle NULLs or date formats). The case study (Table 4) is especially illustrative.

---

> ### Author Rebuttal · Authors · 2025-07-30
>
> **Dear Reviewer Hp1m,**
>
> Thank you for your thoughtful feedback.
> We sincerely appreciate your recognition of PARROT, and we take your concerns, such as failure analysis and schema handling, seriously.
> We hope our responses help to address them and welcome any further questions or suggestions.
>
> ---
>
> **▶ Q1. The case study is strong, but a taxonomy of common failure modes (e.g., date parsing, alias scope errors, aggregate translation) with frequencies could offer deeper insights.**
>
> Thanks for the insightful suggestion.
> To deepen our analysis, we have added a systematic taxonomy of failure modes based on real-world ByteHouse workload results, along with the error frequency distributions across different LLMs.
> The identified failure cases fall into six main categories, which capture the types of errors raised in your question (e.g., "date parsing" refers to *Syntax Parsing*, and "alias scope errors" refers to *Identifier Resolution*).
>
> - **(1) Syntax Parsing:** SQL statements that violate grammar rules, e.g., missing clauses or invalid keyword usage.
>
> - **(2) Identifier Resolution:** References to undefined or ambiguous database objects, e.g., using a column name when multiple tables share the same field.
>
> - **(3) Function Resolution:** Use of unknown or misclassified functions, e.g., calling a non-existent function or incorrectly treating a scalar function as an aggregate.
>
> - **(4) Function Usage:** Valid functions called with incorrect argument types, counts, or values, e.g., passing a string where a number is expected.
>
> - **(5) Type Compatibility:** Operations that combine incompatible data types or schemas, e.g., performing UNION ALL between two queries with different returned value types.
>
> - **(6) Other Errors:** Miscellaneous issues outside the above categories, such as unsupported SQL features or incomplete syntax constructs.
>
> |          **Mode**         | **DeepSeek-R1 32B** | **DeepSeek-V3 671B** | **DeepSeek-R1 671B** | **GPT-4o** | **o3-mini** | **o1-preview** | **Claude 3.7 Sonet** | **Doubao 1.5 Pro** | **Doubao 1.5 Pro Thinking** |
> |:-------------------------|---------------------|----------------------|----------------------|------------|-------------|----------------|----------------------|--------------------|-----------------------------|
> |     **Syntax Parsing**    |         0.05        |       **0.28**       |         0.10         |    0.00    |     0.11    |      0.03      |         0.03         |        0.14        |             0.01            |
> | **Identifier Resolution** |         0.02        |         0.00         |         0.07         |    0.00    |     0.01    |      0.06      |         0.00         |        0.01        |             0.00            |
> |  **Function Resolution**  |       **0.28**      |         0.00         |         0.05         |  **0.40**  |   **0.64**  |    **0.37**    |         0.04         |      **0.27**      |           **0.20**          |
> |     **Function Usage**    |       **0.62**      |       **0.72**       |       **0.70**       |  **0.60**  |   **0.24**  |    **0.54**    |       **0.93**       |      **0.58**      |           **0.79**          |
> |   **Type Compatibility**  |         0.01        |         0.00         |         0.08         |    0.00    |     0.00    |      0.00      |         0.00         |        0.00        |             0.00            |
> |      **Other Errors**     |         0.02        |         0.00         |         0.00         |    0.00    |     0.00    |      0.00      |         0.00         |        0.00        |             0.00            |
>
> We observed two key findings:
> - **(1) Failure patterns vary across LLMs.**
> For example, DeepSeek-V3 671B mostly fails on *Function Usage (0.72)* and *Syntax Parsing (0.28)*, indicating difficulties in correctly applying functions and adhering to SQL grammar.
> In contrast, o3-preview struggles more with *Function Resolution (0.37)* and *Function Usage (0.54)*, suggesting challenges in identifying correct functions and using them properly.
> - **(2) Most errors arise from built-in function misuse.**
> *Function Resolution* and *Function Usage* account for 89\% of errors, highlighting the complexity of translating functions correctly across SQL dialects.
>
> These findings align with the complexity posed by the diverse and extensive range of custom functions implemented across databases, which significantly increases the difficulty for LLMs to accurately identify, and translate functions between dialects.
>
> We will include this breakdown and a visualized summary in our paper to provide deeper insight into model performance.
>
> ---
>
> **▶ Q2. Although the dataset is used to evaluate many LLMs, the authors provide limited discussion on how prompt engineering or decoding strategies impact results.**
>
> Thanks for your valuable comment.
> We have conducted additional experiments to assess the impact of different prompt engineering and decoding strategies.
>
> Specifically, we assessed three types of prompts:
>
> - **(1) Instruction Only:** Simple task instructions specifying dialect translation.
> - **(2) In-Context Learning (ICL-based):** Few-shot demonstrations with example translated SQL pairs.
> - **(3) PARROT:** The comprehensive prompt with detailed Illustrations used in our main experiments (see Table 7 in the Appendix).
>
> For decoding strategies, we varied the top-p parameter across values such as 0.1, 0.5, and 1.0, ranging from more deterministic generation (focusing on high-probability tokens) to increasingly diverse and stochastic outputs through nucleus sampling.
>
> |             **Model**             | **$Acc\_{EX}$** | **$Acc\_{RES}$** |
> |:---------------------------------|:------------:|:-------------:|
> | **GPT-4o (Instruction Only)**  |     23.32    |     20.12     |
> | **GPT-4o (ICL-based)**         |     20.41    |     18.66     |
> | **GPT-4o (PARROT)**            |   **23.91**  |   **21.87**   |
> | **o3-mini (Instruction Only)** |     55.39    |     51.31     |
> | **o3-mini (ICL-based)**        |     57.43    |     52.77     |
> | **o3-mini (PARROT)**           |   **58.60**  |   **54.23**   |
>
> |             **Model**             | **$Acc\_{EX}$** | **$Acc\_{RES}$** |
> |:---------------------------------|:------------:|:-------------:|
> | **GPT-4o (top_p = 0.1)**   |     23.03    |     20.41     |
> | **GPT-4o (top_p = 0.5)** |   22.16  |   18.08   |
> | **GPT-4o (top_p = 1.0)**   |   **24.20**  |     **22.74**     |
>
> As shown in the table above, these variations result in only minor differences in overall accuracy, within a 2\% to 4\% range.
> This limited impact is likely because SQL-to-SQL translation is a highly structured task with strict correctness criteria, i.e., the output must precisely match the target syntax, data types, and built-in functions.
> Unlike open-ended language generation, there is little room for variation or creativity, so changes in prompt design or decoding parameters have minimal effect on accuracy.
>
> We will include these results in the paper and provide all necessary details in our code repository for reproducibility.
>
> ---
>
> **▶ Q3. The $Acc\_{RES}$ metric assumes fully aligned schemas, but the paper lacks details on how schema differences, type coercion, or NULL handling are managed. In practice, cross-system translation often involves schema mismatches and differences in NULL semantics. Clarifying these aspects would improve reproducibility.**
>
> Thank you for your insightful comments.
> The handling of schema differences, type coercion, and NULL semantics within the $Acc\_{RES}$ metric involves three key aspects.
>
> - First, we leverage commercial tools (e.g., Navicat) to migrate source databases to target systems, including DDL schema definitions and underlying data.
> This process also involves manual adjustments, such as aligning data types and verifying data completeness to ensure schema consistency.
> - Second, we require LLMs to explicitly address differences in type coercion and NULL handling during translation.
> For instance, as illustrated in our case study in Table 4, the LLM must convert "virtual\_T1.day" to the appropriate string type and correctly handle the potential NULL values when translating from PostgreSQL to ClickHouse SQL.
> - Finally, we relax the $Acc\_{RES}$ result consistency criteria to accommodate minor variations, such as numeric precision differences and unordered results when no ORDER BY clause is present, considering implementation discrepancies across databases.
> Additionally, the metric supports customization to include or exclude such minor variations based on specific user requirements.
>
> We will clarify these procedures and their implications in the paper to improve reproducibility and transparency.

---

> > ### Comment · Reviewer_Hp1m · 2025-08-05
> >
> > I appreciate the authors' efforts, and I believe most of the issues have been addressed. However, I found the authors lacking a review of some recent NL2SQL methods. I believe a subsection titled 'NL2SQL methods/solutions' should be added to "Related Work", which can provide a more comprehensive background on the research. The following is a non-exhaustive list:\
> > [1] SQL-R1: Training Natural Language to SQL Reasoning Model By Reinforcement Learning\
> > [2] SQL-o1: A Self-Reward Heuristic Dynamic Search Method for Text-to-SQL\
> > [3] ROUTE: Robust Multitask Tuning and Collaboration for Text-to-SQL\
> > ...
> >
> > Given my previous negative rating, I will raise the score accordingly.

---

> > > ### Author Response · Authors · 2025-08-05
> > > **Thanks for your efforts**
> > >
> > > Thank you for your valuable suggestion. We will incorporate a discussion of these recent NL2SQL methods into Section 6 (Related Work) of our paper to provide more comprehensive background for our study.

---

> ### Author Response · Authors · 2025-08-05
> **Follow-Up on Response & Open Discussion**
>
> **Dear Reviewer Hp1m,**
>
> Thank you once again for your thorough review and constructive feedback on our paper. We sincerely appreciate the time and effort you dedicated to evaluating our work.
>
> In response to your comments, we’ve made the following updates:
> - **(1) Failure Analysis:** Included a detailed taxonomy of common failure modes to better characterize limitations of different LLMs **(for Limitation 1)**.
> - **(2) Experimental Validation:** Conducted experiments analyzing both prompt engineering and decoding strategy impacts **(for Limitation 2)**.
> - **(3) Methodology Clarification:** Expanded our explanation of handling fully aligned schemas for better reproducibility **(for Limitation 3)**.
>
> **As we approach the end of the discussion period, we'd be grateful to know if our responses adequately address your concerns or if you'd like any additional clarification.** We're committed to ensuring all concerns are fully resolved and we’re happy to provide further clarification if anything would be helpful.
>
> Thank you again for your expertise to strengthen our paper. We truly appreciate your contributions to improving our work.
>
> **Best regards,**
>
> **The PARROT Authors.**

---

### Official Review · Reviewer_tkLc · 2025-07-04

**Rating:** 5
**Confidence:** 4

**Summary:**

This paper focuses on the task of SQL-to-SQL translation across database systems, which aims to convert a SQL query written for one database system (e.g., PostgreSQL) into a semantically equivalent and syntactically compatible query for another system (e.g., ClickHouse). This task holds substantial practical value in real-world applications—such as system migration, cross-platform interoperability, and query federation—but has been largely underexplored in prior research.
The core contributions can be summarized as:
1. the PARROT Benchmark: a SQL-to-SQL translation benchmark that spans 22 different production-grade database systems, addressing a long-overlooked yet practically important task;
2. Benchmark Construction Methodology: SQL queries are collected from 38 open-source benchmarks as well as real-world workloads from Bytedance. Three dataset variants are constructed: PARROT (598 query pairs), PARROT-Diverse (28003 pairs), PARROT-Simple (5306 pairs);
3. Evaluation metrics: They propose two execution-first metrics to evaluate SQL translation quality: AccEX, which measures the executability of translated queries on the target system, and AccRES, which assesses semantic correctness by comparing result equivalence between source and target executions. These two metrics jointly capture semantic correctness and syntactic fidelity, providing a practical evaluation metrics for cross-system SQL translation;
4. Empirical Analysis: Evaluation of multiple open-source and commercial large language models (LLMs) reveals suboptimal performance on this benchmark, with accuracies ranging from 17% to 60%, underscoring the difficulty of the task. Moreover, authors analyzes model performance sensitivity with respect to SQL query length, dialect pairs, and model size.

**Additional Feedback:**

The rebuttal strengthens the paper. Clarifying that Acc_{RES} compares full result sets over real databases rather than a few synthetic rows is important; please bake these specifics into §4 with per-dialect dataset descriptors (tables/row counts, result-equivalence tolerances for float/decimal, collation/NULL semantics, ordering guarantees) so that Acc_{EX}/Acc_{RES} are unambiguous and fully reproducible (see metric definitions already in §4: Acc_{EX} executability; Acc_{RES} strict result identity). Beyond the helpful failure taxonomy, consider publishing a feature-aligned error schema (functions, types, windowing, timezones, implicit cast rules) and mapping each item in PARROT/PARROT-Diverse/PARROT-Simple to those facets to enable per-capability leaderboards. Also: (i) add stratified splits and report stratified significance (bootstrap CIs across queries and dialects); (ii) release the evaluation harness with pinned seeds/versions and a "fail-fast" safety layer (read-only/SELECT-only execution + static guards); (iii) document the anonymization pipeline and contamination checks; (iv) release prompts/scripts for testcase generation used in semantic checking, even if the training rows themselves remain private; and (v) add guidance for multi-hop/bi-directional evaluation as optional tracks. Finally, since PARROT's core sets span 598 curated pairs (main), 28,003 diverse pairs, and 5,306 unit-style snippets across 22 systems, it would help to expose per-dialect coverage dashboards to show where headroom remains.

**Dataset Code Accessibility:**

Yes

**Dataset Code Comments:**

Authors has already updated the dataset/benchmark and codebase urls onto the OpenReview.
Dataset url: https://huggingface.co/datasets/weizhoudb/PARROT
Codebase url: https://github.com/weAIDB/PARROT

**Ethical Considerations:**

No, there are no or only very minor ethics concerns

**Final Justification:**

I maintain Accept (5). The paper targets a timely, practically consequential task — cross-system SQL translation — that existing NL2SQL benchmarks cannot probe due to limited dialect diversity and system-specific phenomena. The benchmark composition (PARROT core + Diverse + Simple) is clearly described, with scope and scale appropriate for NeurIPS Datasets & Benchmarks, and the evaluation protocol is concrete (execution-first Acc_{EX}/Acc_{RES} with reference executors and schema normalization). Empirically, the paper documents sizeable performance gaps for strong models in both curated cross-dialect pairs and a real ByteHouse workload — evidence that the task is far from solved and that the resource will drive research. The rebuttal addresses my main concerns: it clarifies semantics evaluation at scale, adds a concrete failure taxonomy, expands baselines (RAG/SFT/hybrid with SQLGlot), and commits to language fixes and discussion of safety/robustness. Provided these clarifications (and the taxonomy visualization) are incorporated in the camera-ready, and the evaluation harness/prompts are released alongside the already announced leaderboard, the artifact will be high-leverage for both academic and industrial communities. My remaining suggestions (stratified significance; feature-aligned capability tags; explicit tolerance rules) are straightforward to integrate and will further increase diagnostic value. Overall, the contribution is well-motivated, carefully engineered, and demonstrably impactful; I recommend acceptance.

**Limitations Weaknesses:**

1. Limitations of Semantic Equivalence Evaluation: The AccRES metric relies on a small number of automatically generated test cases to assess result consistency, which may lead to limited coverage and potential false positives or false negatives;
2. Lack of Systematic Error Analysis: While this paper includes some case studies, it lacks a taxonomy of failure modes (e.g., incorrect translation of temporal functions, inconsistent NULL handling), which would help in diagnosing model weaknesses;
3. Insufficient Discussion on Safety and Bias: This paper does not address potential safety, robustness, or bias issues that may arise from automated SQL translation across systems;
4. Language Quality Issues: This paper contains some stylistic and grammatical problems, including typographical errors (e.g., “calcluation”) and awkward phrasings (e.g., “the translation errors can be classified into two categories, tightly coupled with the characteristics...”). A thorough language revision is recommended to improve clarity and academic rigor;
5. Limited Baseline Scope: Although this paper includes both open-source and proprietary LLMs, it only evaluates prompt-based inference;
6. Restricted Evaluation Scope: This benchmark focuses solely on pairwise SQL-to-SQL translation and does not cover more complex scenarios such as multi-hop translation (e.g., PostgreSQL-to-Oracle-to-SQL Server) or bi-directional translation tasks.

**Strengths Contributions:**

1. Real-World Relevance: This paper identifies and systematically studies cross-database SQL translation—an important yet underexplored task frequently encountered in enterprise environments;
2. Data Construction Pipeline: The PARROT benchmark is constructed through collection, anonymization, deduplication, clustering, annotation, and error-guided filtering. The inclusion of real-world production data ensures both diversity and quality;
3. Evaluation Metrics: The proposed metrics—AccEX (executability) and AccRES (result consistency)—offer a more meaningful assessment than traditional string-matching metrics, capturing both syntactic correctness and semantic fidelity;
4. Empirical Analysis: This paper provides detailed evaluations across various LLMs, database systems, and SQL lengths, revealing the current limitations of LLMs in this setting and offering valuable insights for future research.

---

> ### Author Rebuttal · Authors · 2025-07-30
>
> **Dear Reviewer tkLc,**
>
> Thank you for the thoughtful and detailed review.
> We truly appreciate your recognition of PARROT as well as your constructive suggestions, such as refining semantic evaluation and failure analysis.
> We hope our responses address your concerns and welcome any further questions or feedback.
>
> ---
>
> **▶ Q1. Limitations of Semantic Equivalence Evaluation: The $Acc\_{RES}$ metric relies on a small number of automatically generated test cases to assess result consistency, which may lead to limited coverage and potential false positives or false negatives.**
>
> Thanks for the insightful comment.
>
> - First, we emphasize that $Acc\_{RES}$ does not rely on a small number of automatically generated test cases.
> Instead, it evaluates result consistency using full-scale benchmark databases (from Bytedance and public sources), which contain substantial data.
> For example, the BIRD datasets used in our evaluation have an average of 549K rows.
> - Second, to further reduce noise and avoid potential false positives or negatives, we exclude source queries that return empty or tiny result sets unless they are aggregated results, ensuring more meaningful and reliable comparisons.
>
> We will clarify these details in the paper.
>
> ---
>
> **▶ Q2. Lack of Systematic Error Analysis: While this paper includes some case studies, it lacks a taxonomy of failure modes (e.g., incorrect translation of temporal functions, inconsistent NULL handling), which would help in diagnosing model weaknesses.**
>
> Thanks for the valuable suggestion.
>
> We have added a detailed failure mode analysis based on the ByteHouse real-world workload results, including error frequency distributions across different LLMs.
> We organize failures into six categories, which capture the types of errors raised in your question (e.g., incorrect translation of temporal functions falls into *Function Usage*).
>
> - **(1) Syntax Parsing:** SQL statements that violate grammar rules, e.g., missing clauses or invalid keyword usage.
> - **(2) Identifier Resolution:** References to undefined or ambiguous database objects, e.g., using a column name when multiple tables share the same field.
> - **(3) Function Resolution:** Use of unknown or misclassified functions, e.g., calling a non-existent function or incorrectly treating a scalar function as an aggregate.
> - **(4) Function Usage:** Valid functions called with incorrect argument types, counts, or values, e.g., passing a string where a number is expected.
> - **(5) Type Compatibility:** Operations that combine incompatible data types or schemas, e.g., performing UNION ALL between two queries with different returned value types.
> - **(6) Other Errors:** Miscellaneous issues outside the above categories, such as unsupported SQL features or incomplete syntax constructs.
>
> |          **Mode**         | **DeepSeek-R1 32B** | **DeepSeek-V3 671B** | **DeepSeek-R1 671B** | **GPT-4o** | **o3-mini** | **o1-preview** | **Claude 3.7 Sonet** | **Doubao 1.5 Pro** | **Doubao 1.5 Pro Thinking** |
> |:-------------------------|---------------------|----------------------|----------------------|------------|-------------|----------------|----------------------|--------------------|-----------------------------|
> |     **Syntax Parsing**    |         0.05        |       **0.28**       |         0.10         |    0.00    |     0.11    |      0.03      |         0.03         |        0.14        |             0.01            |
> | **Identifier Resolution** |         0.02        |         0.00         |         0.07         |    0.00    |     0.01    |      0.06      |         0.00         |        0.01        |             0.00            |
> |  **Function Resolution**  |       **0.28**      |         0.00         |         0.05         |  **0.40**  |   **0.64**  |    **0.37**    |         0.04         |      **0.27**      |           **0.20**          |
> |     **Function Usage**    |       **0.62**      |       **0.72**       |       **0.70**       |  **0.60**  |   **0.24**  |    **0.54**    |       **0.93**       |      **0.58**      |           **0.79**          |
> |   **Type Compatibility**  |         0.01        |         0.00         |         0.08         |    0.00    |     0.00    |      0.00      |         0.00         |        0.00        |             0.00            |
> |      **Other Errors**     |         0.02        |         0.00         |         0.00         |    0.00    |     0.00    |      0.00      |         0.00         |        0.00        |             0.00            |
>
> From the table above, we observe:
> - **(1) Failure patterns vary across LLMs.**
> For example, DeepSeek-V3 671B mostly fails on *Function Usage (0.72)* and *Syntax Parsing (0.28)*, indicating difficulties in correctly applying functions and adhering to SQL grammar.
> In contrast, o3-preview struggles more with *Function Resolution (0.37)* and *Function Usage (0.54)*, suggesting challenges in identifying correct functions and using them properly.
> - **(2) Most errors arise from built-in functions.**
> *Function Resolution* and *Function Usage* account for 89\% of errors, highlighting challenges with dialect-specific functions.
>
> These observations align with the complexity posed by the diverse and extensive range of custom functions implemented across databases, which significantly increases the difficulty for LLMs to accurately identify, and translate functions between dialects.
>
> We will include this taxonomy and a visual summary in our paper to provide a more systematic understanding of model weaknesses.
>
> ---
>
> **▶ Q3. Insufficient Discussion on Safety and Bias: This paper does not address potential safety, robustness, or bias issues that may arise from automated SQL translation across systems.**
>
> Thanks for raising this critical point.
> - First, our benchmark is designed to enable systematic evaluation of robustness and bias by including a diverse and balanced set of SQL dialects.
> This allows us to assess how model performance varies across systems with differing syntax, semantics, and function sets.
> To further support fair evaluation, we can incorporate metrics during calculation of $Acc_{EX}$ and $Acc_{RES}$ that weigh samples by dialect level, ensuring more representative scoring.
> - Second, regarding safety concerns, external static analysis tools can be integrated to flag potentially risky queries (e.g., those involving data deletion or modification), and evaluations by DBA experts or LLM-based evaluators (e.g., LLM-as-a-Judge techniques) can further assess whether any unsafe operations or sensitive information are introduced.
>
> We will include a discussion of these aspects in our paper.
>
> ---
>
> **▶ Q4. Language Quality Issues: This paper contains some stylistic and grammatical problems, including typographical errors (e.g., “calcluation”) and awkward phrasings (e.g., “the translation errors can be classified into two categories, tightly coupled with the characteristics...”). A thorough language revision is recommended to improve clarity and academic rigor.**
>
> Thanks for pointing this out.
> We have thoroughly reviewed the paper and made corrections to grammar, spelling, and awkward phrasing to improve clarity and maintain academic rigor.
>
> ---
>
> **▶ Q5. Limited Baseline Scope: Although this paper includes both open-source and proprietary LLMs, it only evaluates prompt-based inference.**
>
> Thanks for the helpful comment.
> To expand the baseline scope, we will include:
> - (1) RAG-based methods that incorporate similar translation pairs from PARROT-DIVERSE into the prompt;
> - (2) SFT-based methods trained on instruction-style SQL pairs from PARROT-DIVERSE;
> - (3) Hybrid methods that combine LLMs with rule-based methods (SQLGlot, which supports 30 SQL dialects).
> Below, we present the results of the hybrid methods, where SQLGlot provides an intermediate translation further refined by LLM.
>
> |               **Model**               | **$Acc\_{EX}$** | **$Acc\_{RES}$** |
> |:---------------------------------|:------------:|:-------------:|
> | **SQLGlot**                       |     36.59    |     31.05     |
> | **DeepSeek-V3 671B**              |     39.94    |     32.65     |
> | **DeepSeek-V3 671B (+ SQLGlot)** |     41.40    |     34.11     |
> | **DeepSeek-R1 671B**              |     46.94    |     40.52     |
> | **DeepSeek-R1 671B (+ SQLGlot)** |     47.52    |     40.82     |
> | **o3-mini**                       |     58.60    |     54.23     |
> | **o3-mini (+ SQLGlot)**          |     59.18    |     54.81     |
> | **o1-preview**                    |     56.26    |     48.69     |
> | **o1-preview (+ SQLGlot)**       |     58.60    |     50.44     |
>
> As shown in the table above, this approach yields only modest improvements (average +2.40%) on the real-world ByteHouse workload (requiring manual translation by expert DBAs).
>
> These results underscore the difficulty of our benchmark, where even enhanced methods struggle to generalize.
> Achieving better performance requires more sophisticated strategies, such as query decomposition and targeted integration of rule-based and LLMs.
>
> ---
>
> **▶ Q6. Restricted Evaluation Scope: This benchmark focuses solely on pairwise SQL-to-SQL translation and does not cover more complex scenarios such as multi-hop translation (e.g., PostgreSQL-to-Oracle-to-SQL Server) or bi-directional translation tasks.**
>
> Thank you for the thoughtful comment.
>
> - First, we note that pairwise SQL-to-SQL translation reflects the majority of real-world scenarios, such as cross-system migrations which typically involve a single source and target dialect.
> - Second, our benchmark is flexible and extensible. It can support more complex settings:
>   * **Multi-hop translation** can be simulated by chaining pairwise samples that share the same source query but differ in targets.
>   * **Bi-directional translation** can be evaluated by simply reversing the source-target pairs.
>
> We will include a discussion of these extended use cases in our paper to clarify the broader applicability of our benchmark.

---

> ### Author Response · Authors · 2025-08-07
> **Follow-Up on Response & Open Discussion**
>
> **Dear Reviewer tkLc,**
>
> Thank you once again for your thorough review and constructive feedback. We sincerely appreciate the time and care you have taken to evaluate our work.
>
> In response to your concerns, we have made the following updates:
> - **(1) Semantic Equivalence Evaluation:** Added evaluation details for handling limited coverage (via original database migration and empty or small-result exclusion) **(for Limitation 1)**.
> - **(2) Systematic Error Analysis:** Introduced a failure-mode taxonomy as well as a detailed analysis to better characterize weaknesses of different LLMs **(for Limitation 2)**.
> - **(3) Safety and Bias Discussion:** Clarified PARROT’s bias and robustness evaluation (balanced and diverse SQL dialects) and safety measures (external tools, DBA experts, and LLM-based evaluators) **(for Limitation 3)**.
> - **(4) Language Quality Issue:** Revised the paper thoroughly to improve grammar, clarity, and academic rigor **(for Limitation 4)**.
> - **(5) Baseline Scope:** Expanded comparisons to other methods such as hybrid (rule-based + LLM) ones **(for Limitation 5)**.
> - **(6) Evaluation Scope:** Added explanations for supporting complex scenarios (pairwise samples with shared SQL intermediates) **(for Limitation 6)**.
>
> **As we approach the end of the discussion period, we would be grateful to know if our responses have adequately addressed your concerns or if you would like any further clarifications.** Your feedback has been valuable in strengthening our work, and we are committed to resolving any remaining questions.
>
> Thank you again for your expertise to strengthen our paper. We truly appreciate your contributions to improving our work.
>
> **Best regards,**
>
> **The PARROT Authors.**

---

### Official Review · Reviewer_hPXw · 2025-07-06

**Rating:** 5
**Confidence:** 5

**Summary:**

**Summary**
1. This paper introduces PARROT (Practical And Realistic Benchmark for Cross System SQL Translation), a new benchmark designed specifically for evaluating the ability of LLMs to perform cross-system SQL to SQL translation.

2. While there are multiple prior works that explore the task of Natural Language to SQL generation the authors highlight that the task of SQL to SQL translation also significant practical utility for instance in the event of migrating an SQL query created for one database system such as MySQL into an equivalent query for another database system such as ClickHouse. The authors argue that existing SQL benchmarks are ill-suited for this task because they typically focus on a single system and fail to capture the complex, system-specific dialect differences (e.g. unique functions, data types, syntax rules) found in real-world scenarios.

3. To address this gap, the authors curate a diverse collection of real-world SQL queries from 38 open-source benchmarks and private enterprise workloads. This paper details the rigorous curation workflow employed to create a challenging and realistic dataset. Furthermore the author conduct experiments on the PARROT benchmark that suggest that LLMs like GPT-4o, Claude Sonnet 3.7, DeepSeek R1 etc struggle with this task, highlighting significant room for improvement.

**Contributions**
1. The authors release the PARROT benchmark with 3 variants
   - PARROT: 598 challenging, manually verified translation pairs from diverse, real-world sources.
   - PARROT-Diverse: 28,003 pairs across 22 database systems for extensive syntax and dialect testing.
   - PARROT-Simple: 5,306 simple, representative samples for focused stress-testing of specific translation rules.

2. The author's provide a rigorous data curation methodology including a 5 step workflow for creating the benchmark which includes
   - Collecting real-world SQL queries.
   - Standardizing and anonymizing data.
   - Filtering for complexity and diversity.
   - Using an ensemble of rule-based tools and LLMs for annotation.
   - Performing error-guided selection to ensure correctness.

3. The author's perform a experimental comparative analysis of open-sourced and closed LLMs on the benchmark and highlight the significant room for improvement on the benchmark

4. The authors release PARROT, along with its variants, source code, and a public leaderboard to facilitate future research.

**Dataset Code Accessibility:**

Yes

**Dataset Code Comments:**

The dataset code is available under here: https://github.com/weAIDB/PARROT
The code is well written and the dataset is available to download via hugging-face cli.

The command/script/instructions to trigger the automatic-evalauation on the benchmark however seems to be missing. The authors should update the README with this command and the code with the script.

**Ethical Comments:**

No ethical concerns

**Ethical Considerations:**

No, there are no or only very minor ethics concerns

**Final Justification:**

The authors have provided additional analysis that was requested. This is a technically solid paper with extensive analysis now. Hence i retain my ratings as accept

**Limitations Weaknesses:**

1. An in-depth qualitative analysis of failure cases is lacking in the paper. It would be good to find common patterns of failure in modern day LLMs and present them in the Appendix for completeness
2. As an additional optional analysis the author's could include results of Gemini 2.5 pro in the comparison as it ranks number 1 on LMSYS for code.
3. As a part of the queries are derived from the public benchmarks it is possible that some LLMs may have been trained on the data. Therefore there is a risk that the benchmarks might be skewed towards measuring memorization in LLMs. Additional steps may be needed to transform the queries to be sufficiently distinct while retaining the semantics so as to avoid this.
4. The efficiency of the translated queries compared to the original queries is not measured in the benchmark. For instance in an edge case scenario the translated query might be much slower on the new DB system or may timeout or there might be other ways of rewriting the query to be more efficient. Having this additional signal would also provide a measure of quality of translation

**Strengths Contributions:**

1. The benchmark is novel and the first to introduce SQL to SQL translation task (as stated by authors and to the best of my knowledge). Current existing benchmarks such as BIRD have only looked at the Text to SQL translation tasks.
2. The benchmark offers practical utility as database migrations are common for enterprises and the task of automating the translation of SQL queries is particularly relevant in these cases. The benchmark also highlights the significant room for improvement in modern day LLMs for this task.
3. The paper is very well written. The data curation methodology is rigorous and methodical. The experimentation analysis with different open and closed source models is also well presented and sound.

Overall the paper is extremely strong and makes solid contributions to the field.

---

> ### Author Rebuttal · Authors · 2025-07-30
>
> **Dear Reviewer hPXw,**
>
> Thank you for your thoughtful and encouraging review.
> We sincerely appreciate your recognition of PARROT and your valuable suggestions regarding failure cases, potential memorization, and query efficiency.
> We hope our responses are helpful, and we’d be happy to respond to any further questions you may have.
>
> ---
>
> **▶ Q1. An in-depth qualitative analysis of failure cases is lacking in the paper. It would be good to find common patterns of failure in modern day LLMs and present them in the Appendix for completeness.**
>
> Thanks for your thoughtful suggestion.
> We have added a detailed analysis of failure cases based on the results in real-world ByteHouse workloads, including error frequency distributions across different LLMs.
> These failures fall into six categories.
> - **(1) Syntax Parsing:** SQL statements that violate grammar rules, e.g., missing clauses or invalid keyword usage.
> - **(2) Identifier Resolution:** References to undefined or ambiguous database objects, e.g., using a column name when multiple tables share the same field.
> - **(3) Function Resolution:** Use of unknown or misclassified functions, e.g., calling a non-existent function or incorrectly treating a scalar function as an aggregate.
> - **(4) Function Usage:** Valid functions called with incorrect argument types, counts, or values, e.g., passing a string where a number is expected.
> - **(5) Type Compatibility:** Operations that combine incompatible data types or schemas, e.g., performing UNION ALL between two queries with different returned value types.
> - **(6) Other Errors:** Miscellaneous issues outside the above categories, such as unsupported SQL features or incomplete syntax constructs.
>
> |          **Model**         | **DeepSeek-R1 32B** | **DeepSeek-V3 671B** | **DeepSeek-R1 671B** | **GPT-4o** | **o3-mini** | **o1-preview** | **Claude 3.7 Sonet** | **Doubao 1.5 Pro** | **Doubao 1.5 Pro Thinking** |
> |:-------------------------|---------------------|----------------------|----------------------|------------|-------------|----------------|----------------------|--------------------|-----------------------------|
> |     **Syntax Parsing**    |         0.05        |       **0.28**       |         0.10         |    0.00    |     0.11    |      0.03      |         0.03         |        0.14        |             0.01            |
> | **Identifier Resolution** |         0.02        |         0.00         |         0.07         |    0.00    |     0.01    |      0.06      |         0.00         |        0.01        |             0.00            |
> |  **Function Resolution**  |       **0.28**      |         0.00         |         0.05         |  **0.40**  |   **0.64**  |    **0.37**    |         0.04         |      **0.27**      |           **0.20**          |
> |     **Function Usage**    |       **0.62**      |       **0.72**       |       **0.70**       |  **0.60**  |   **0.24**  |    **0.54**    |       **0.93**       |      **0.58**      |           **0.79**          |
> |   **Type Compatibility**  |         0.01        |         0.00         |         0.08         |    0.00    |     0.00    |      0.00      |         0.00         |        0.00        |             0.00            |
> |      **Other Errors**     |         0.02        |         0.00         |         0.00         |    0.00    |     0.00    |      0.00      |         0.00         |        0.00        |             0.00            |
>
> From the table above, we have two observations.
> - **(1) Failure patterns vary across different LLMs.**
> For instance, DeepSeek-V3 671B primarily struggles with *Function Usage (0.72)* and *Syntax Parsing (0.28)*, indicating difficulties in correctly applying functions and adhering to SQL grammar.
> In contrast, o1-preview exhibits a higher rate of failures in *Function Resolution (0.37)* and *Function Usage (0.54)*, suggesting challenges in identifying correct functions and using them properly.
> - **(2) The majority of failures stem from misuse of built-in functions.**
> For example, *Function Resolution* and *Function Usage* account for 89\% of all errors on average, highlighting that the translation of database-specific functions is the most significant obstacle.
>
> These observations align with the complexity posed by the diverse and extensive range of custom functions implemented across databases, which significantly increases the difficulty for LLMs to accurately identify, and translate functions between dialects.
>
> We will include this table (or a visualized figure) along with the detailed breakdown in the appendix to improve the completeness.
>
> ---
>
> **▶ Q2. As an additional optional analysis the author's could include results of Gemini 2.5 pro in the comparison as it ranks number 1 on LMSYS for code.**
>
> Thanks for the suggestion.
> We have added the results of Gemini 2.5 Pro, where "$\rightarrow$ PostgreSQL" indicates that the target dialect for the translation is PostgreSQL.
>
> |       **Model**       | **$\rightarrow$ PostgreSQL** | **$\rightarrow$ MySQL** | **$\rightarrow$ Oracle** | **$\rightarrow$ DuckDB** | **$\rightarrow$ SQL Server** |
> |:---------------------|:-------------------------------------:|:------------------------------:|:-------------------------------:|:-------------------------------:|:-----------------------------------:|
> | **Gemini 2.5 pro**    |                 48.28                 |              23.53             |              32.76              |              53.57              |                21.05                |
> | **DeepSeek R1 671B**  |                 48.28                 |              44.12             |              50.00              |              42.86              |                36.84                |
> | **GPT-4o**            |               **58.62**               |              **50.00**             |              55.17              |            **60.71**            |              **42.11**              |
> | **Claude 3.7 Sonnet** |               **58.62**               |              44.12             |            **58.00**            |              42.86              |                36.84                |
>
> As shown in the table above, Gemini 2.5 Pro exhibits performance variation across dialects, i.e., achieving 50.93\% average accuracy on PostgreSQL and DuckDB, but only 22.29\% on MySQL and SQL Server.
> This reflects the uniqueness and challenge of the SQL-to-SQL task, which demands not only general language understanding but also fine-grained mastery of dialect-specific syntax and semantics that current models, including top-ranked ones, still struggle with.
>
> We will include these results and the analysis in Section 5.
>
> ---
>
> **▶ Q3. As a part of the queries are derived from the public benchmarks it is possible that some LLMs may have been trained on the data. Therefore there is a risk that the benchmarks might be skewed towards measuring memorization in LLMs. Additional steps may be needed to transform the queries to be sufficiently distinct while retaining the semantics so as to avoid this.**
>
> Thanks for raising this critical point.
>
> - First, we can process queries from public benchmarks  using the same three levels of anonymization described in Section 3.2 (e.g., replacing table and column names), which helps ensure they remain semantically meaningful yet distinct from their original forms.
> - Second, although some queries are adapted from public benchmarks, they are initially written in a single dialect, and their equivalents in the target dialects are labeled by us. The absence of labeled SQL pairs minimizes the risk of test data leakage.
>
> ---
>
> **▶ Q4. The efficiency of the translated queries compared to the original queries is not measured in the benchmark. For instance in an edge case scenario the translated query might be much slower on the new DB system or may timeout or there might be other ways of rewriting the query to be more efficient. Having this additional signal would also provide a measure of quality of translation.**
>
> Thanks for the suggestion.
> We have included an efficiency analysis by comparing execution time ratios between translated target and original source queries, measured as $\frac{Time_{target}}{Time_{source}}$ on the real-world ByteHouse workload in PARROT.
>
> |          **Model**          | **Mean** | **Median** | **95th** | **99th** |
> |:---------------------------|:--------:|:----------:|:--------:|:--------:|
> | **DeepSeek-R1 32B**         |   0.59   |  **0.49**  |   1.18   |   1.32   |
> | **DeepSeek-V3 671B**        |   0.62   |    0.53    |   1.13   | **1.25** |
> | **DeepSeek-R1 671B**        |   0.58   |    0.51    | **1.11** |   1.38   |
> | **GPT-4o**                  |   0.63   |    0.50    |   1.33   |   1.44   |
> | **o3-mini**                 |   0.59   |    0.53    |   1.12   |   1.38   |
> | **o1-preview**              | **0.57** |    0.50    |   1.15   |   1.37   |
> | **Claude 3.7 Sonet**        |   0.62   |    0.51    |   1.20   |   1.33   |
> | **Doubao 1.5 Pro**          |   0.60   |  **0.49**  |   1.26   |   1.33   |
> | **Doubao 1.5 Pro Thinking** |   0.58   |    0.50    |   1.18   |   1.37   |
>
> As shown in the table above, the translated queries do not exhibit significantly degraded performance (e.g., no extreme slowdowns or timeouts).
>
> We will add this efficiency analysis and discussion to the paper.
>
> ---
>
> **▶ Q5. The command/script/instructions to trigger the automatic-evalauation on the benchmark however seems to be missing. The authors should update the README with this command and the code with the script.**
>
> Thanks for the helpful comment.
> We will update the `validator` directory in our code repository to include the automatic evaluation script along with clear instructions in the README for triggering the evaluation.
> The repository will be updated as soon as this year's policy allows.

---

> > ### Comment · Reviewer_hPXw · 2025-08-04
> >
> > I thank the authors for providing the additional analysis for the camera ready paper. I retain my ratings as accept

---

> > > ### Author Response · Authors · 2025-08-04
> > > **Thanks for your recognition**
> > >
> > > We are truly grateful for your kind recognition of our work and the time you have taken to share your thoughtful feedback with us.

---

### Official Review · Reviewer_BoF9 · 2025-07-08

**Rating:** 5
**Confidence:** 3

**Summary:**

This paper introduces PARROT, a new and comprehensive benchmark designed to evaluate LLMs in cross-system SQL translation (SQL-to-SQL translation), adapting a SQL query written for one database system (e.g., MySQL) into an equivalent query for another (e.g., ClickHouse). PARROT consists of 598 manually verified translation pairs from 38 open-source benchmarks and real-world business services. It also offers extensive variants like PARROT-Diverse (28,003 pairs) and PARROT-Simple (5,306 pairs), covering 22 production-grade database systems for extensive syntax and focused stress testing. Empirical analysis using PARROT reveals that state-of-the-art LLMs achieve low accuracy, underscoring significant room for future research in this domain.

**Additional Feedback:**

- Figure 1 has two portions of SQLite.

**Dataset Code Accessibility:**

Yes

**Dataset Code Comments:**

The dataset and code are clearly documented and provided.

**Ethical Considerations:**

No, there are no or only very minor ethics concerns

**Final Justification:**

The authors have provided reasonable responses to my comments and additional experiments addressing my concerns.

**Limitations Weaknesses:**

- Does query migration happen frequently? If so, is it better to use text-to-SQL for new database systems instead of SQL-to-SQL? Discussing this would strengthen the argument more effectively.
- What is the performance of rule-based methods for this task. How does combining rule-based approaches with LLMs perform? This would more effectively gauge the benchmark's difficulty.

**Strengths Contributions:**

- The paper is clearly written and easy to follow.
- The benchmark is the first large-scale benchmark dedicated to cross-system SQL translation (SQL-to-SQL).
- The benchmark construction process is rigorous and well-designed.

---

> ### Author Rebuttal · Authors · 2025-07-30
>
> **Dear Reviewer BoF9,**
>
> Thank you for your thoughtful review and recognition of PARROT.
> We appreciate the valuable suggestions, including the frequency of query migration and the performance of rule-based methods, which help to strengthen our discussion and evaluation.
> We sincerely hope our responses address your concerns and welcome any further questions you may have.
>
> ---
>
> **▶ Q1. Does query migration happen frequently? If so, is it better to use text-to-SQL for new database systems instead of SQL-to-SQL? Discussing this would strengthen the argument more effectively.**
>
> Thanks for your thoughtful comments.
>
> **1. Query migration is a frequent, real-world necessity.**
>
> Query migration is essential in numerous routine tasks, including:
>
> - (1) Cross-database analytics tools that continuously translate SQL between heterogeneous systems to enable seamless, unified analysis;
> - (2) Extract-Transform-Load (ETL) workflows, where SQL is used extensively to extract data from source systems, transform it through filtering, joining, or aggregation, and then load it into target databases with potentially different dialects.
>
> For instance, at ByteDance, auto-generated queries from business intelligence tools require translation when migrating from Hive or Spark to ClickHouse database to ensure performance and compatibility.
> This process sustains a production workload exceeding 1,000 queries per second (QPS) while adapting syntax, optimizations, and engine-specific functions.
>
> **2. Reasons to focus on SQL‑to‑SQL rather than text‑to‑SQL.**
>
> SQL-to-SQL is a more fundamental task that can serve as a foundation for a wide range of applications (e.g., data transformation pipelines, object-relational mapping tools, and cross-database interoperability), including text-to-SQL by translating their output (often tailored for SQLite) automatically into production‑ready dialects used across different enterprise systems.
> In contrast, text-to-SQL is primarily designed for natural language interfaces and faces both technical challenges (e.g., semantic ambiguity, limited coverage of complex SQL constructs) and application constraints.
>
> Moreover, we note that other reviewers have also emphasized the significance of SQL‑to‑SQL translation.
>
> - **Reviewer hPXw:**
> > "The benchmark offers practical utility as database migrations are common for enterprises and the task of automating the translation of SQL queries is particularly relevant in these cases."
>
> - **Reviewer tkLc:**
> > "This task holds substantial practical value in real-world applications—such as system migration, cross-platform interoperability, and query federation—but has been largely underexplored in prior research."
>
> - **Reviewer Hp1m:**
> > "The paper addresses a practically important yet underexplored task—cross-dialect SQL translation—which is critical for enterprise-level database migration, system interoperability, and real-world automation."
>
> We will revise the paper to highlight the practical significance of this task, broader applicability, and how it complements existing research directions.
>
> ---
>
> **▶ Q2. What is the performance of rule-based methods for this task. How does combining rule-based approaches with LLMs perform? This would more effectively gauge the benchmark's difficulty.**
>
> Thanks for your helpful comments.
>
> **1. Performance of rule-based methods.**
>
> Rule-based methods are typically designed for particular dialects, e.g., Ora2Pg supports only translations from Oracle and MySQL to PostgreSQL, and relies heavily on hand-crafted rules.
> Extending them to handle diverse dialects requires significant manual effort by expert DBAs to identify equivalent constructs and encode translation rules, leading to variation of translation accuracy across dialects.
>
> **2. Empirical study of hybrid methods.**
>
> We conducted additional experiments combining a rule-based tool (i.e., SQLGlot, which supports 30 SQL dialects and has 8.1k GitHub stars) with LLMs.
> In this hybrid setup, SQLGlot’s output serves as an intermediate translation result within the LLM workflow.
>
> |               **Model**               | **$Acc\_{EX}$** | **$Acc\_{RES}$** |
> |:---------------------------------|:------------:|:-------------:|
> | **SQLGlot**                       |     36.59    |     31.05     |
> | **DeepSeek-V3 671B**              |     39.94    |     32.65     |
> | **DeepSeek-V3 671B (+ SQLGlot)** |     41.40    |     34.11     |
> | **DeepSeek-R1 671B**              |     46.94    |     40.52     |
> | **DeepSeek-R1 671B (+ SQLGlot)** |     47.52    |     40.82     |
> | **o3-mini**                       |     58.60    |     54.23     |
> | **o3-mini (+ SQLGlot)**          |     59.18    |     54.81     |
> | **o1-preview**                    |     56.26    |     48.69     |
> | **o1-preview (+ SQLGlot)**       |     58.60    |     50.44     |
>
> As shown in the table above, combining SQLGlot with LLMs yields only modest gains (average +2.40%) on the real-world ByteHouse workload.
> This highlights the difficulty of our benchmark, where even hybrid approaches struggle to generalize.
> Improved performance still requires non-trivial strategies, such as segmenting queries and selectively applying rule-based and LLM methods.
>
> We will add discussions about rule-based methods and the experiment results in our paper.
>
> ---
>
> **▶ Q3. Figure 1 has two portions of SQLite.**
>
> Thanks for your comments.
> - First, the two SQLite portions in Figure 1 are independent: one represents benchmarks involving a single dialect (SQLite only), and the other represents those involving two dialects, one of which is SQLite.
> - Second, we will clarify this in the caption and visually distinguish the circles (i.e., highlight the boundary of each circle) to make it easier to understand.

---

> > ### Comment · Reviewer_BoF9 · 2025-08-05
> > **Official Comment by Reviewer BoF9**
> >
> > I thank the authors for providing additional experiments and reasonable responses to my comments. I have raised my score.

---

> > > ### Author Response · Authors · 2025-08-05
> > > **Thanks for your efforts**
> > >
> > > We sincerely appreciate your time and thoughtful review of our work. Thank you for recognizing our efforts in addressing your comments and for your constructive feedback throughout this process.

---

> ### Author Response · Authors · 2025-08-05
> **Follow-Up on Response & Open Discussion**
>
> **Dear Reviewer BoF9,**
>
> Thank you once again for your thoughtful review and valuable feedback on our paper. We sincerely appreciate the time and effort you dedicated to evaluating our work.
>
> In response to your comments,
> - (1) We have clarified the frequency of query migration **(for Limitation 1)**.
> - (2) We have included performance comparisons with rule-based methods **(for Limitation 2)**.
> - (3) We have provided an explanation for the two portions of SQLite in Figure 1 **(for Additional Feedback)**.
>
> **As we approach the end of the discussion period, we would like to inquiry if you have any further questions or concerns regarding our responses.** We’re more than happy to provide any additional clarification if needed.
>
> Thank you again for your insightful review. We truly appreciate your contributions to improving our work.
>
> **Best regards,**
>
> **The PARROT Authors.**

---

### Note · Authors · 2025-08-12

**Dear Chairs and Reviewers,**

We sincerely appreciate the time you have devoted to evaluating our work. Your thoughtful feedback and constructive suggestions have been valuable in strengthening our paper, and we are grateful for your recognition of our contributions.
- **Reviewer BoF9:**
> "the first large-scale benchmark dedicated to cross-system SQL translation ... construction process is rigorous and well-designed"
- **Reviewer hPXw:**
> "the paper is extremely strong and makes solid contributions to the field"
- **Reviewer tkLc:**
> "systematically studies cross-database SQL translation—an important yet underexplored task"
- **Reviewer Hp1m:**
> "addresses a practically important yet underexplored task"

We are pleased and grateful that our responses have addressed your concerns and strengthened the paper in the rebuttal phase.
Below, we summarize the key updates we will incorporate into the final version:

**1. Failure Analysis**

We have added a failure-mode taxonomy as well as a detailed analysis of different LLMs *(Reviewer hPXw-Q1, Reviewer tkLc-Q2, Reviewer Hp1m-Q1)*

**2. Baseline Scope**
- Added Gemini 2.5 pro results *(Reviewer hPXw-Q2)* and efficiency evaluations *(Reviewer hPXw-Q4)*
- Compared hybrid methods (rule-based+LLM) *(Reviewer BoF9-Q2, Reviewer tkLc-Q5)*
- Assessed different prompts and decoding strategies *(Reviewer Hp1m-Q2)*

**3. Equivalence Evaluation**

We have clarified evaluation details for handling limited coverage *(Reviewer tkLc-Q1)* and fully aligned schemas *(Reviewer Hp1m-Q3)*

**4. Benchmark Expansion**
- Clarified how open-source SQLs are transformed to mitigate LLM memorization concerns *(Reviewer hPXw-Q3)*
- Included safety and bias measurements *(Reviewer tkLc-Q3)* and extended support for complex scenarios *(Reviewer tkLc-Q6)*

Additionally, we sincerely appreciate Reviewer tkLc’s updated feedback to:
- Clarify $Acc_{RES}$ with per-dialect dataset descriptors
- Introduce per-capability leaderboards based on failure taxonomy
- Refine our repository with additional instructions including dataset splits, and complex scenario evaluations

We will also discuss recent NL2SQL methods in Section 6 *(Reviewer Hp1m)*.

Once again, we extend our deepest gratitude for your insightful feedback, which has been valuable in strengthening the paper's rigor, clarity, and overall impact. We sincerely appreciate the time and expertise you have contributed to this review process.

**Best regards,**

**The PARROT Authors.**

---

### Decision · Program_Chairs · 2025-09-18

**Decision:**

Accept (poster)

**Comment:**

This paper introduces PARROT, a novel and comprehensive benchmark for Cross-System SQL Translation (SQL-to-SQL). The authors correctly identify that while Text-to-SQL has been extensively studied, the practical and frequent task of migrating SQL queries between different database systems (e.g., from MySQL to ClickHouse) remains under-explored. Existing benchmarks are ill-suited for this task as they typically focus on a single dialect (like SQLite) and fail to capture the rich, system-specific functions, data types, and syntax rules found in production environments.
The main contribution is the PARROT benchmark suite, which includes three variants of the datasets. The authors propose two execution-based metrics, AccEX (executability) and AccRES (result equivalence), to provide a robust evaluation framework. Their empirical analysis reveals that even state-of-the-art Large Language Models (LLMs) perform poorly on PARROT, with average accuracy below 38.53%, highlighting the benchmark's difficulty and the significant room for future research.

All reviewers recognizing the novelty and practical importance of the proposed benchmark. The reviewers praised the rigorous and well-designed data curation methodology, the paper's clarity, and its potential impact on the field.
There are several aspects where the paper can be further improved, including expansion of baselines and scope, deeper failure analysis, and the reviewers recognized the additional efforts made by the authors during rebuttal.